



# Intact polar lipids in the hadal seabed of the Atacama Trench point to lateral sediment transport and *in situ* production as key sources of labile organic matter

Edgart Flores[1,2,3]*, Sebastian I. Cantarero[4], Paula Ruiz-Fernández[1,2,3], Nadia Dildar[4], Matthias Zabel[5], Osvaldo Ulloa[2,3] and Julio Sepúlveda[3,4]*

[1]Programa de Postgrado en Oceanografía, Departamento de Oceanografía, Facultad de Ciencias Naturales y Oceanográficas, Universidad de Concepción, Concepción, Chile
[2]Departamento de Oceanografía, Universidad de Concepción, Casilla 160-C, Concepción, Chile
[3]Millennium Institute of Oceanography, Universidad de Concepción, Concepción, Chile
[4]Department of Geological Sciences and Institute of Arctic and Alpine Research, University of Colorado Boulder, Boulder, CO 80309, USA
[5]MARUM – Center for Marine Environmental Sciences & Department of Geosciences, Univ. of Bremen, D-28334, Bremen, Germany

*Correspondence to:* Edgart Flores (edgart.flores@imo-chile.cl), Julio Sepúlveda (jsepulveda@colorado.edu)

**Abstract.** Elevated concentrations of organic matter are found in sediments of hadal trenches relative to those found in the abyssal seabed, but the origin of such biological material remains elusive. Here, we report the composition and distribution of cell membrane intact polar lipids (IPLs) in surface sediments around the deepest points of the Atacama Trench and adjacent bathyal depths to assess and constrain the sources of labile organic matter in the hadal seabed. Multiscale bootstrap resampling of IPLs' structural diversity and abundance indicates distinct lipid signatures in the sediments of the Atacama Trench that are more closely related to those found in bathyal sediments than to those previously reported for the upper ocean water column in the region. While the overall number of unique IPL structures in hadal sediments is limited and they contribute a small fraction of the total IPL pool, they include a high contribution of phospholipids with mono- and di-unsaturated fatty acids that are not associated with photoautotrophic sources. The diversity of labile IPLs in hadal sediments of the Atacama Trench suggests the presence of *in situ* microbial production and biomass that resembles traits of physiological adaptation to high pressure and low temperature, and/or the transport of labile organic matter from shallower sediment. We argue that the export of the most labile lipid component of the organic matter pool from the euphotic zone and the overlying oxygen minimum zone into the hadal sediments is neglectable. Our results contribute to the understanding of the mechanisms that control the delivery of labile organic matter to this extreme deep-sea ecosystem, whereas they provide insights into some potential physiological adaptation of the *in situ* microbial community to high pressure and low temperature through lipid remodeling.

## 1. Introduction

The deep ocean has been classically considered a vast "biological desert" (Danovaro et al., 2003) due to the attenuation of organic matter fluxes with increasing ocean depth (Wakeham et al., 1984; Martin et al., 1987;



Hedges et al., 2001; Rex et al., 2006). However, hadal trenches (~6,000-11,000 m below sea level) contradict this
long-held paradigm (Danovaro et al., 2003; Glud et al., 2013; Leduc et al., 2016; Wenzhöfer et al., 2016; Luo et
al., 2017) as they act as depocenters of organic matter (Jahnke and Jahnke, 2000) and hotspots for microbial
activity (Glud et al., 2013; Wenzhöfer et al., 2016; Liu et al., 2019). In hadal trench systems, while hydrostatic
pressure appears to be the most important physical factor controlling biological activity compared to shallower
habitats (Jamieson et al., 2010; Tamburini et al., 2013), the availability of organic matter has been considered the
major factor controlling the abundance, biomass, and diversity of life in the deep ocean (Danovaro et al., 2003;
Ichino et al., 2015). However, our knowledge on the composition and potential sources of organic matter in marine
trenches remains limited. A study by Xu et al. (2018) indicates that the main sources of organic matter to the hadal
zone are: (1) the vertical sinking of particulate organic matter; (2) the carrion falls of dead bodies; (3) inputs of
terrestrial organic matter; (4) lateral transport of organic matter from continental slopes under the combined effects
of trench topography and gravity (Jahnke et al., 1990; Fischer et al., 2009; Inthorn et al., 2006; Ichino et al., 2015),
or by earthquakes (Glud et al., 2013; Kioka et al., 2019); and (5) *in situ* chemosynthetic production associated
with cold seeps or hydrothermal vents. Previous studies have highlighted the importance of particulate organic
matter sinking mainly from the euphotic zone (Stockton and DeLaca, 1982; Angel, 1984; Gooday et al., 2010). In
fact, the settling flux of particulate matter measured by sediment traps at 4,000 m of depth in the North Pacific
Subtropical Gyre Ocean reveals that a seasonal export pulse can exceed the mean annual flux by ~150% (Poff et
al., 2021). However, it is unknown whether such pulses reach the hadal sediments (6,000-11,000 m) with the same
intensity and impact. Independent of the main sources of organic matter, which can vary spatially and temporally,
the channeling of allochthonous organic matter to the hadal zone should be facilitated by the characteristic V-
shape cross-section of trenches (Itou et al., 2000; Itoh et al., 2011; Bao et al., 2018). Additionally, autochthonous
sources of organic matter associated with *in situ* bacterial and archaeal biomass production may exist (Smith,
2012; Nunoura et al., 2016; Ta et al., 2019; Hand et al., 2020), but their overall contribution as a secondary input
to carbon budgets and energy flow in these systems remains poorly constrained. The spatial differences seen in
benthic prokaryotic populations in hadal regions such as the Mariana, Japan, and Izu-Ogasawara trenches have
been attributed to the variability of biogeochemical conditions, mainly nitrate and oxygen (Hiraoka et al., 2020),
with the latter showing large variability in its consumption (Glud et al., 2021). Furthermore, the abundance of
prokaryotes in hadal depths can be influenced by sediment redistribution (Schauberger et al., 2021), which in turn
may be influenced by the intensity of propagating internal tides (Turnewitsch et al., 2014). All these factors likely
alter the deposition, distribution, and composition of organic matter present in trench sediments.

Recent metagenomic studies have revealed the presence of abundant heterotrophic microorganisms in sediments
of the Challenger Deep (Nunoura et al., 2018), which are likely fueled by the endogenous recycling of available
organic matter (Nunoura et al., 2015; Tarn et al., 2016). Although less specific than genomic markers, cell
membrane intact polar lipids (IPLs) allow more quantitative estimates of microbial biomass in nature (e.g., Lipp
et al., 2008; Schubotz et al., 2009; Cantarero et al., 2020). IPLs are composed of a polar head group typically
attached to a glycerol backbone from which aliphatic chains are attached via ester and/or ether bonds (Sturt et al.,
2004). Their structural diversity is given by the modifications found in the different components of their chemical
structure (e.g., polar head groups can be comprised of phosphorous, nitrogen, sulfur, sugars, and amino acids),
whereas aliphatic chains (alkyl or isoprenoidal) can vary in their length (number of carbon atoms), and their degree
of unsaturation, methylation, hydroxylation, and cyclization (Van Mooy and Fredricks, 2010; Brandsma et al.,



2011; Schubotz et al., 2013). In bacteria and eukarya, alkyl chains are most commonly linked via an ester bond to
the sn-glycerol-3-phosphate backbone (Koga and Morii, 2007), although some bacteria are known to produce di-
and tetraether lipids (Weijers et al., 2007). Archaea produce membranes rich in di- and tetraether isoprenoidal
lipids (Volkman et al., 1989; Pearson and Ingalls, 2013), where isoprenoids are linked via an ether bond to the *sn*-
glycerol-1-phosphate backbone (Jain et al., 2014). The variability of membrane chemical structures underlies the
adaptability of microbial lifestyles to changing environmental conditions such as nutrients, temperature, oxygen,
pH, and pressure (DeLong and Yayanos, 1985; Somero, 1992; Van Mooy et al., 2009; Carini et al., 2015;
Sebastián et al., 2016; Siliakus et al., 2017; Boyer et al., 2020). Furthermore, since eukaryotic and bacterial ester-
bond IPLs are more labile than ether-bond counterparts (Logemann et al., 2011), they are suitable biomarkers to
evaluate sources of labile organic matter in marine environments.

IPLs have been previously used as microbial markers in diverse marine settings, such as along strong redox
gradients in the Black Sea (Schubotz et al., 2009b) and the oxygen minimum zones (OMZs) of the eastern tropical
Pacific (Schubotz et al., 2018a; Cantarero et al., 2020) and the Arabian Sea (Pitcher, 2011), as well as in surface
open ocean waters of the eastern south Pacific (Van Mooy and Fredricks, 2010), the northwestern Atlantic
(Popendorf et al., 2011b), and the Mediterranean Sea (Popendorf et al., 2011a), to name a few. Their utility as
markers of microbial diversity and processes has also been tested in marine sediments (Liu et al., 2011, 2012;
Sturt et al., 2004), such as along the Peru Margin, Equatorial Pacific, Hydrate Ridge, and Juan de Fuca Ridge
(Lipp and Hinrichs, 2009a) and in subsurface sediment layers from the Peru Margin (Biddle et al., 2006).
However, to the best of our knowledge, no IPL studies have been reported for sediments of hadal trenches.
In this study, we investigate the chemical diversity and abundance of microbial IPLs as markers of labile organic
matter sources in the deepest points of the Atacama Trench sediments (AT), and compare them to similar IPL
stocks in shallower surface sediments (~500-1,200 m) and in the overlying water column (upper 700 m; Cantarero
et al., 2020). More specifically, we evaluate possible organic matter provenance (*in situ* vs. allochthonous
production) and some potential chemical adaptations of the *in situ* microbial community to the extreme conditions
of the hadal region.

## 2. Material and Methods

### 2.1. Study areas and sampling

The AT is located in the eastern tropical South Pacific (ETSP) along the Peru-Chile margin, and it underlies the
eutrophic and highly productive Humboldt Current System (Angel, 1982; Ahumada, 1989), which includes the
fourth largest (by volume) oxygen minimum zone (OMZ) in the world (Schneider et al., 2006). In this area, while
there is minimal river runoff (Houston, 2006), winds can transfer dust from the adjacent continental desert (Angel,
1982). With an extension of ~5,900 km, the AT is the world's largest trench (Sabbatini et al., 2002), whereas it is
geographically isolated from other trenches in the Pacific Ocean.

In this study, we investigated the diversity and abundance of IPLs in a total of 9 hadal surface sediments (3 sites
between 7,734 and 8,063 m water depth, with 3 depth intervals each) collected during the HADES-SO261 cruise
(March to April 2018) onboard the RV *Sonne* (Wenzhöfer, 2019) , and 7 bathyal surface sediments (7 sites; 529-
1200 m water depth) collected during the ChiMeBo-SO211 cruise (November 2-29, 2010) onboard the RV *Sonne*





(Matys et al., 2017) (Table 1; Fig. 1). We compare our results against IPL results from the overlying water column
(0-700 m) recently reported in Cantarero et al. (2020).


**Table 1. Sampling stations from the Hades, ChiMeBo, and LowpHOX-2 expeditions.**


| Cruise-RV | Device | Enviroment | Station | Environmental samples | Sampling depth (m) | Latitude (°S) | Longitude (°w) | Date | Reference |
|---|---|---|---|---|---|---|---|---|---|
| HADES SONNE SO-261 | Multi--corer (MUC) | Hadal sediments | A10 | Hadal sediments (0-1, 1-2 and 2-3 cm) | 7734 | 20.32 | 71.29 | 26/03/2018 | |
| | | | A4 | Hadal sediments (0-1, 1-2 and 2-3 cm) | 7890 | 23.81 | 71.37 | 11/03/2018 | This study |
| | | | A5 | Hadal sediments (0-1, 1-2 and 2-3 cm) | 8063 | 23.36 | 71.34 | 14/03/2018 | |
| ChiMeBo SONNE SO-211 | Multi--corer (MUC) | Bathyal Sediments | B12 | Upper bathyal sediment (0-1 cm) | 529 | 23.59 | 70.67 | 02-29/11/2010 | |
| | | | B08 | Upper bathyal sediment (0-1 cm) | 539 | 25.2 | 70.68 | 02-29/11/2010 | |
| | | | B22 | Upper bathyal sediment (0-1 cm) | 545 | 27.29 | 71.05 | 02-29/11/2010 | |
| | | | B07 | Lower bathyal sediment (0-1 cm) | 920 | 25.07 | 70.66 | 02-29/11/2010 | This study |
| | | | B05 | Lower bathyal sediment (0-1 cm) | 957 | 27.5 | 71.13 | 02-29/11/2010 | |
| | | | B11 | Lower bathyal sediment (0-1 cm) | 1113 | 23.85 | 70.65 | 02-29/11/2010 | |
| | | | B04 | Lower bathyal sediment (0-1 cm) | 1200 | 27.45 | 71.16 | 02-29/11/2010 | |
| LowpHOX-2 Cabo de Hornos | Rosette (Niskin bottles) | Water column | T3/T5 | Chlorophyll maximun (0.3-2.7 μm) | 9-10 | 20.07/20.03 | 70.36/70.89 | 04-06/02/2018 | |
| | | | T3/T5 | Upper chemocline (0.3-2.7 μm) | 25-28 | 20.07/20.03 | 70.36/70.89 | 04-06/02/2018 | |
| | | | T3/T5 | Lower chomocline (0.3-2.7 μm) | 35-45 | 20.07/20.03 | 70.36/70.89 | 04-06/02/2018 | Cantarero et al., 2020 |
| | | | T3/T5 | Upper OMZ (0.3-2.7 μm) | 55-60 | 20.07/20.03 | 70.36/70.89 | 04-06/02/2018 | |
| | | | T3/T5 | Core OMZ (0.3-2.7 μm) | 250 | 20.07/20.03 | 70.36/70.89 | 04-06/02/2018 | |
| | | | T3/T5 | Mesopelagic zone (0.3-2.7 μm) | 750 | 20.07/20.03 | 70.36/70.89 | 04-06/02/2018 | |


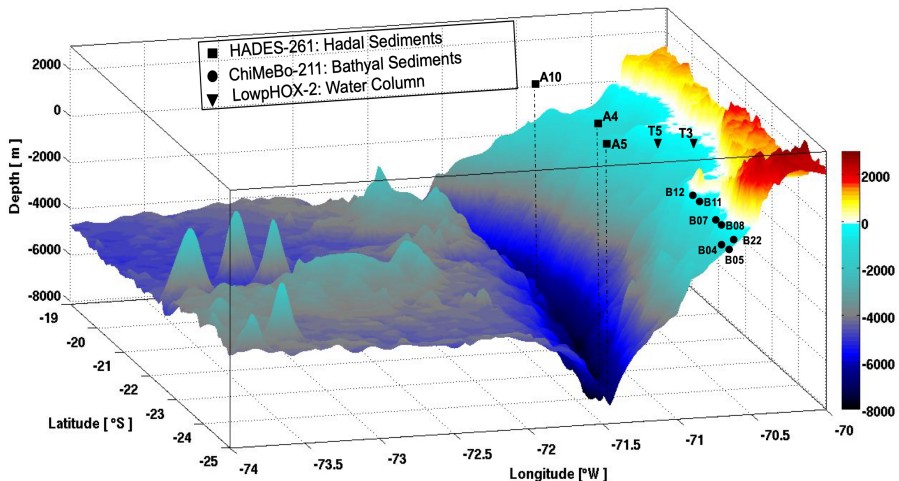

**Figure 1. Three-dimensional map of the Atacama Trench showing the sampling locations of this study. The black**
**squares indicate the hadal sediment sampling stations, the black circles indicate the bathyal sediment sampling stations**
**from Matys et al. (2017), and the black triangles indicate water column sampling stations from Cantarero et al. (2020).**



Surface sediments from the bathyal (0-1 cm) and the hadal AT region (0-1 cm, 1-2 cm and 2-3 cm) were collected
using a multi-corer (MUC) equipped with twelve 60-cm-long acrylic tubes (6-10 cm diameter for bathyal
sediments and 9.5 cm diameter for hadal sediments). During the HADES expedition, an autonomous lander
equipped with a Seabird SBE-19 plus CTD and 2 Niskin bottles (30 L) was used to obtain hydrographic data down
to ~7850 m (SO261 cruise). Hadal sediments from the HADES-SO261 cruise were cold-stored after the sampling,
and during their transit to the University of Bremen and then to the University of Colorado Boulder. Information
about the sampling of bathyal sediments during the ChiMeBo-SO211 cruise can be found in Matys et al. (2017).
All samples were processed, extracted, and analyzed in the Organic Geochemistry Laboratory at the University
of Colorado Boulder as described below.

We compare our IPL results from surface sediment in the hadal and bathyal regions against samples from the
overlying water column from the LowPhOx-2 cruise recently reported by Cantarero et al. (2020). This includes
size-fractionated suspended organic matter (0.3-2.7 µm and 2.7-53 µm) at two stations and from six water depths
that are representative of the dominant biogeochemical zonation associated with the OMZ of this region:
chlorophyll maximum (~10 m), upper chemocline (~25 m), lower chemocline (~45 m), upper OMZ (~60 m), core
OMZ (~250 m), and mesopelagic zone (~750 m) (See Table 1 and Cantarero et al., 2020 for further details).

**2.2. Analytical methods**
**2.2.1 Lipid extraction**
Sediment samples from the AT were freeze dried before extraction. Approximately 1-2 grams of dry sediment
was placed in a combusted glass centrifuge tube and extracted using a modified version (Wörmer et al., 2013) of
the Bligh and Dyer Extraction method (Bligh and Dyer, 1959) as detailed in Cantarero et al. (2020). Briefly, before
extraction, we added 1 µg of C16 PAF ($C_{26}H_{54}NO_7P$) to each sample as an internal standard. Samples were
sequentially   extracted   using   Dichloromethane:MeOH:Phosphate   buffer   (1:2:0.8   v:v:v;   2x),
Dichloromethane:MeOH:Trichloroacetic buffer (1:2:0.8 v:v:v; 2x) and Dichloromethane:MeOH (1:5 v:v; 1x).
After each addition, samples were vortexed for 30 seconds, sonicated for 10 minutes, and then centrifuged for 5
minutes at 2000 rpm. Each extraction was then transferred to a separatory funnel where a total lipid extract (TLE)
was combined and then concentrated under a gentle $N_2$ stream. Before analysis, the TLEs were resuspended in
Dichloromethane:Methanol (9:1) v/v and filtered through a 0.45 µm polytetrafluoroethylene (PTFE) syringe filter.
The processing and extraction of bathyal sediments from the ChiMeBo-SO211 cruise and water column samples
from the LowpHOx-2 cruise has been reported by Matys et al. (2017) and Cantarero et al. (2020), respectively.
TLEs were transferred into 2 ml vials with 200 µl inserts, and dissolved in 100 µl of Dichloromethane:MeOH
[9:1, v:v].

**2.2.2. IPL analysis**
IPL were analyzed according to Wörmer et al. (2013) and as described in Cantarero et al. (2020) using a Thermo
Scientific Ultimate 3000 High Performance Liquid Chromatograph (HPLC) coupled to a Q Exactive Focus
Orbitrap-Quadrupole High Resolution Mass Spectrometer (HPLC-HRMS) via electrospray ionization (ESI). The
HPLC program comprised a flow rate of 0.4 mL/min using a mixture of two mobile phases: mixture A consisted





of Acetonitrile:Dichloromethane (75:25, v:v) with 0.01% formic acid and 0.01% NH$_4$OH; mixture B consisted of
Methanol:H$_2$O (50:50, v:v) with 0.4% formic acid and 0.4% NH$_4$OH. We used a linear gradient as follows:  1%
B (0–2.5 min), 5% (4 min), 25% B (22.5 min), 40% B (26.5 min–27.5 min), and the HPLC column was kept at
40 °C. Samples were injected (10 µl) dissolved in Dichloromethane:Methanol (9:1, v:v).
ESI settings comprised: sheath gas (N$_2$) pressure 35 (arbitrary units), auxiliary gas (N$_2$) pressure 13 (arbitrary
units), spray voltage 3.5 kV (positive ion ESI), capillary temperature 265°C, S-Lens RF level 55 (arbitrary units).
The instrument was calibrated for mass resolution and accuracy using the Thermo Scientific Pierce LTQ Velos
ESI Positive Ion Calibration Solution (containing a mixture of caffeine, MRFA, Ultramark 1621, and N-
butylamine in an acetonitrile/methanol/acetic acid solution).

IPLs were identified on positive ionization, on both full scan and data depended MS$^2$, based on their molecular
weights as either protonated (M + H)$^+$ or ammonium (M + NH$_4$)$^+$ adducts compounds, fragmentation patterns, and
retention times, and as compared against relevant literature (Sturt et al., 2004; Schubotz et al., 2009a; Wakeham
et al., 2012) and the internal database of the Organic Geochemistry Lab at CU Boulder.

The peak areas of individual IPLs were integrated using the Thermo Fisher Scientific TraceFinder software using
extracted ion chromatograms of their characteristic molecular ions. IPL abundances were determined with a
combination of an internal standard (C$_{16}$PAF, Avanti Lipids) for recovery estimates and an external calibration to
a linear regression between peak areas and known concentrations of an IPL cocktail comprised of 17 different IPL
classes across a 5-point dilution series (0.001–2.5 ng/µl) (see Cantarero et al., 2020). Deuterated standards (Avanti
Lipids: d7-PC, d7-PG, d7-PE and d9-DGTS) were used to correct for potential matrix effects on ionization
efficiency. The relative response factors followed the order: MGDG >DGTS>DGTA >PDME >PME >PG > PC>
PE >SQDG > DGCC > DGDG. Lipids classes were grouped into phospholipids (PG; phosphatidylglycerol, PE;
phosphatidylethanolamine, PC; phosphatidylcholine, and PME/PDME; Phosphatidyl(di)methylethanolamine);
glycolipids    (MG;    Monoglycosyldiacylglycerol,    DG;    Diglycosyldiacylglycerol,    and    SQ;
Sulfoquinovosyldiacylglycerol), Betaine lipids (DGTA; Diacylglyceryl hydroxymethyl-trimethyl-β-alanine,
DGTS; Diacylglyceryl trimethylhomoserine, and DGCC; Diacylglycerylcarboxy-N-hydroxymethyl-choline) and
Other lipids (Gly-Cer; Glycosidic ceramides, PI; phosphatidylinositol, and OL; Ornithine lipids).

**2.3. Statistical analyses**
We used the Bray–Curtis similarity coefficient (Mirzaei et al., 2008) to produce hierarchical clustering of the
abundance of classes and molecules of IPLs, two types of p-values were available: approximately unbiased (AU)
p-value and bootstrap probability (BP) value with the number of bootstrap replications of 10,000 (Suzuki and
Shimodaira, 2006). We performed non-metric Multidimensional Scaling (NMDS) (Warton et al., 2012) to
examine the dissimilarity between the IPLs in each sample. The calculated distances to group centroids were based
on Bray-Curtis dissimilarity from IPLs abundances matrix, and the significance of the associations was determined
by 999 random permutations. Significance tests of the multivariate dissimilarity between groups were made using
Analysis of Similarity (ANOSIM), where complete separation and no separation among groups is suggested by R
= 1 and R = 0, respectively (Clarke and Gorley, 2015). Statistical differences in the numbers of carbon atoms and
double bonds were identified by ANOVA and Tukey's HSD (honestly significant difference) post hoc test. We
used similarity of percentage (SIMPER) analysis to identify the percentage contributions of IPLs which accounted



for > 90% of the similarity within each cluster. The multivariate statistical analyzes, as well as other statistical
analyses were calculated using the Vegan package (Oksanen et al., 2013) of open-source software R version 3.6.2
within the ggplots package (Warnes et al., 2015).

**3. Results**
**3.1 Hydrographic conditions**
The potential temperature-salinity-dissolved oxygen ($\theta$-s-$O_2$) diagrams revealed an oxygenated and well-mixed
water mass occupying the deeper parts of the AT (Fig. S1). However, the upper 1000 m shows variability in
temperature (12-23 °C), salinity (34.4 - 34.8) and oxygen (0.5-267 μM). In the mesopelagic and bathypelagic zone
of AT between 1000 and 4000 m, more stable physical-chemical conditions are apparent (temperature ~ 2.3 °C,
salinity ~34.6, oxygen ~120.6 μM). Below 4000 m, average conditions were characterized by a potential
temperature ~1.8 ° C, salinity ~34.7, and oxygen ~143 μM (Fig. S1). A physical-chemical characterization of the
water column during the ChiMeBo-SO211, LowpHOx-2, and HADES-SO261 cruises has been reported in Matys
et al. (2017), Cantarero et al. (2020) and Vargas et al. (2021), and Fernández-Urruzola et al. (2021), respectively.

**3.2 IPLs in surface sediments of the Atacama trench**
**3.2.1. Distribution of IPL classes by polar head groups**
The 16 sediment samples from bathyal and hadal regions statistically grouped into four clusters based on their
dominant polar head group classes (Fig. 2, chemical structures in Fig. S2). Clusters 1 and 2 had approximately
unbiased (AU) p-values of 91% and 88%, respectively. Cluster 3 had the highest AU p-value of ≥ 97%, whereas
Cluster 4 had the lowest AU p-value of 61%. The cluster analysis revealed a degree of spatial heterogeneity
between bathyal and hadal depths and between the top three centimeters of hadal sediments, which results in the
lack of a clear separation between hadal and bathyal environments. In addition, the 0-1 cm hadal sediments at A4
station were un-clustered, consistent with a distinct distribution pattern of IPL classes.
Cluster 1, composed of only hadal samples from three different stations and depths, included phospholipids as the
most abundant IPL class (Fig. 2). Clusters 2, 3 and 4, composed of mixed bathyal and hadal samples, were mostly
differentiated by changes in the relative abundances of non-phosphorous IPLs including betaine classes. The un-
clustered sample was characterized by the lowest relative abundance of phospholipids and the highest relative
abundance of betaine lipids (especially DGCC).



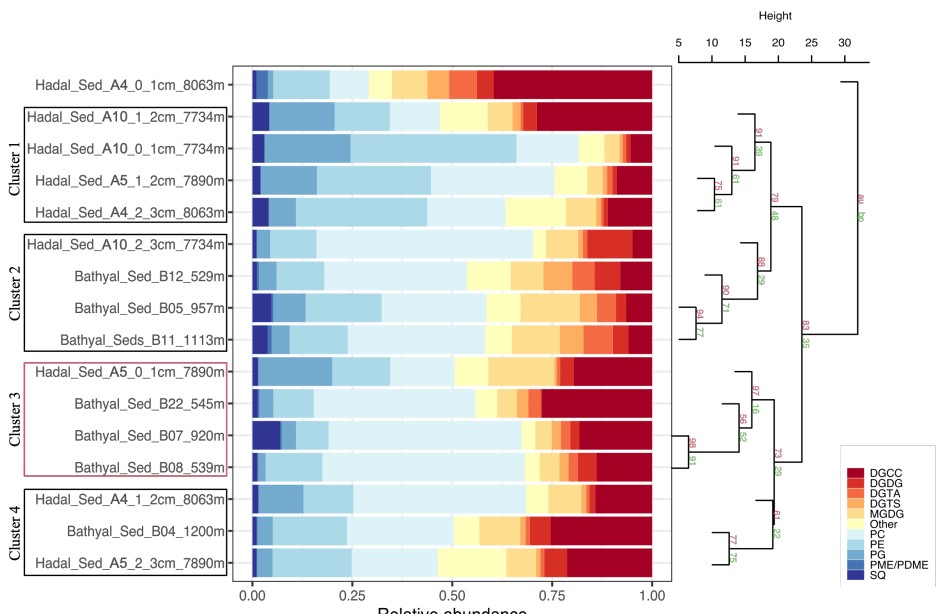

**Figure 2.** Cumulative bar charts of the fractional abundance of IPL classes in each surface sediment sample from the bathyal and hadal regions (left panel). Samples were grouped according to arithmetic mean (UPGMA) hierarchical clustering based on Euclidean distances. The p-values are shown at branches, approximately unbiased (AU) in red and bootstrap probability (BP) in green (right panel). Clusters 3 with an AU ≥ 95% confidence are indicated by the red rectangles (left) and are considered to be strongly supported by the data.

### 3.2.2. Distribution of individual IPLs

An overview of the most important IPLs contributing to dissimilarity between samples (Fig. 3) was obtained through a SIMPER analysis based on Bray-Curtis coefficient within each cluster. Samples in Cluster 1 were on average 59.5% similar, with 14 individual IPLs contributing 50.6% of the total similarity. This cluster exhibited a high contribution of PE (32:1, 33:1, and 34:2), PG (36:2), and DGCC (26:0, 27:0 and 28:0) molecules (Table 2). Additionally, this cluster exhibited a large diversity of PC molecules, with lower relative abundance (Fig. 3). Samples in Cluster 2, on the other hand, which includes mainly bathyal stations, were on average 58.8% similar and exhibited a high contribution of PC (35:0, 32:1, 36:2, 33:1, and 35:1) (Table 2). While this cluster shows a wide range of molecules, including PG, PE and MGDG, their relative contributions are low (Fig. 3). Samples in Cluster 3 were on average 57.3% similar and included three bathyal and one hadal stations. This cluster exhibited a high contribution of DGCC (42:6) and PC (35:0, 33:2, 30:1, and 29:2) molecules (Table 2). Samples in Cluster 4 were on average 63.6% similar, and exhibited a high contribution of PC (30:2, 33:2), DGCC (42:6), MGDG (28:0), and PE (33:2 and 31:2) molecules (Table 2). The un-cluster sample (Hadal sediment of 0-1 cm at A4 station) is mainly composed by the DGCC 42:6 (Fig. 3). In general, phospholipids showed a wide distribution and were found across all sediment samples. The total dissimilarity between Clusters 1 and 2 was 59.17%, with PC 35:0, PE 32:1, PI-AR, PG 36:2, DGCC 27:0, PC 36:2, PC 34:1, PC 32:1, DGCC 26:0, and PC 35:1 contributing 32.4% of it (Table 2). The total dissimilarity between Clusters 1 and 3 was 60.7%, with DGCC 42:6, PC 35:0, PI-AR, PE 32:1, PG 36:2, DGCC 27:0 and 26:0, and PC 33:2 contributing 38.1% of it (Table 2). The total





dissimilarity between Clusters 1 and 4 was 62.5%, with DGCC 42:6, PC 30:2, PE 32:1, PC 35:0, PG 36:2, PC
33:2, and DGCC 27:0 contributing 37.62% of it (Table 2).

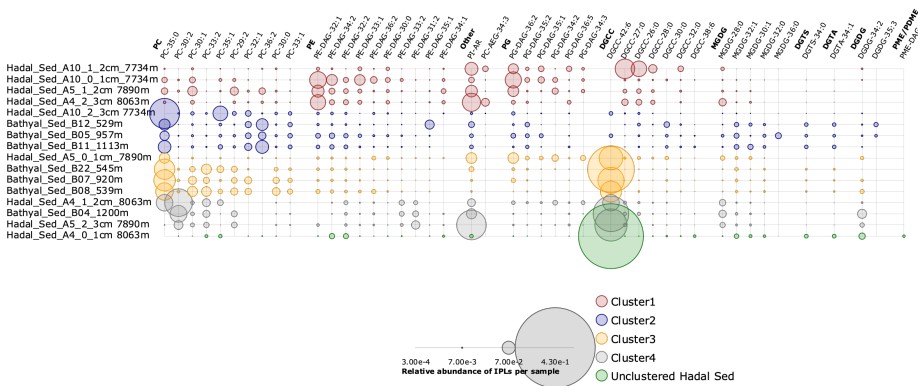

**Figure 3. Relative abundance of individual IPLs contributing most of the dissimilarity between the 4 clusters shown in**
**Fig. 2. Sampling stations were organized from top to bottom and are shown using the same order from hierarchical**
**clusters in Fig. 2, and were organized from left to right by IPL classes. The circle ratios are proportional to the relative**
**abundance of IPLs in each sample (bottom panel).**





**Table 2. SIMPER (similarity percentage) analysis. Average abundance and contribution of IPLs that explain the main differences among the sediment sampling stations is based on the hierarchical clusters shown in Fig. 2.**

### Group Cluster 1 — Cluster 1: Average Similarity = 59.53

| IPLs | Average Cluster 1 | Average Similarity | Similarity/SD | Contribution (%) | Cumulative (%) |
|---|---|---|---|---|---|
| PI-AR | 0.06 | 4.76 | 2.46 | 7.99 | 7.99 |
| PE-DAG-32:1 | 0.06 | 4.37 | 1.45 | 7.34 | 15.33 |
| PG-DAG-36:2 | 0.05 | 3.79 | 2 | 6.36 | 21.69 |
| PE-DAG-33:1 | 0.03 | 2.06 | 33.49 | 3.45 | 25.14 |
| PE-DAG-34:2 | 0.03 | 1.89 | 1.74 | 3.17 | 28.31 |
| DGCC-26:0 | 0.04 | 1.84 | 2.04 | 3.09 | 31.4 |
| PC-30:1 | 0.03 | 1.76 | 2.21 | 2.96 | 34.36 |
| DGCC-27:0 | 0.04 | 1.74 | 1.8 | 2.93 | 37.3 |
| PE-DAG-30:0 | 0.02 | 1.7 | 13.1 | 2.86 | 40.15 |
| PE-DAG-32:2 | 0.02 | 1.39 | 1.07 | 2.34 | 42.49 |
| PC-35:0 | 0.02 | 1.31 | 1.52 | 2.2 | 44.69 |
| DGCC-28:0 | 0.02 | 1.22 | 1.96 | 2.05 | 46.74 |
| PC-26:0 | 0.02 | 1.18 | 1.46 | 1.99 | 48.73 |
| PC-28:0 | 0.02 | 1.14 | 1.59 | 1.91 | 50.63 |

### Group Cluster 2 — Cluster 2: Average Similarity = 58.79

| IPLs | Average Cluster 2 | Average Similarity | Similarity/SD | Contribution (%) | Cumulative (%) |
|---|---|---|---|---|---|
| PC-35:0 | 0.08 | 5.63 | 7.54 | 9.58 | 9.58 |
| PC-32:1 | 0.03 | 3.12 | 31.24 | 5.3 | 14.88 |
| PC-36:2 | 0.05 | 2.74 | 1.13 | 4.67 | 19.55 |
| PC-33:1 | 0.02 | 2.04 | 10.17 | 3.46 | 23.01 |
| PC-35:1 | 0.03 | 1.63 | 4.48 | 2.77 | 25.78 |
| PI-AR | 0.02 | 1.61 | 3.9 | 2.74 | 28.53 |
| MGDG-32:1 | 0.02 | 1.44 | 1.35 | 2.45 | 30.98 |
| PE-DAG-32:1 | 0.02 | 1.38 | 5.03 | 2.35 | 33.33 |
| PE-DAG-34:2 | 0.02 | 1.38 | 2.75 | 2.35 | 35.68 |
| PE-DAG-32:2 | 0.02 | 1.22 | 2.79 | 2.08 | 37.76 |
| PC-32:0 | 0.01 | 1.14 | 5.69 | 1.94 | 39.69 |
| PG-DAG-36:2 | 0.02 | 1.1 | 3.23 | 1.87 | 41.57 |
| PG-DAG-35:2 | 0.02 | 1.09 | 1.23 | 1.86 | 43.43 |
| PC-34:1 | 0.04 | 1.06 | 0.41 | 1.8 | 45.23 |
| PC-30:1 | 0.01 | 1.05 | 7.23 | 1.79 | 47.02 |
| PC-32:2 | 0.01 | 0.95 | 11.7 | 1.61 | 48.64 |
| PC-29:2 | 0.01 | 0.95 | 2.69 | 1.61 | 50.25 |

### Group Cluster 3 — Cluster 3: Average Similarity = 57.31

| IPLs | Average Cluster 3 | Average Similarity | Similarity/SD | Contribution (%) | Cumulative (%) |
|---|---|---|---|---|---|
| DGCC-42:6 | 0.16 | 12.84 | 6.72 | 22.4 | 22.4 |
| PC-35:0 | 0.08 | 4.78 | 1.14 | 8.33 | 30.74 |
| PC-33:2 | 0.03 | 2.07 | 1.19 | 3.61 | 34.35 |
| PC-30:1 | 0.03 | 1.96 | 1.82 | 3.42 | 37.77 |
| PC-29:2 | 0.03 | 1.79 | 1.2 | 3.12 | 40.89 |
| PI-AR | 0.05 | 1.69 | 1.09 | 2.95 | 43.84 |
| MGDG-32:1 | 0.01 | 1.22 | 7.66 | 2.14 | 45.98 |
| PE-DAG-32:1 | 0.01 | 1.18 | 10.45 | 2.05 | 48.03 |
| PC-30:0 | 0.02 | 1.13 | 1.22 | 1.97 | 50 |

### Group Cluster 4 — Cluster 4: Average Similarity = 63.64

| IPLs | Average Cluster 2 | Average Similarity | Similarity/SD | Contribution (%) | Cumulative (%) |
|---|---|---|---|---|---|
| PC-30:2 | 0.12 | 9.04 | | 14.21 | 14.21 |
| DGCC-42:6 | 0.14 | 8.91 | | 13.99 | 28.2 |
| PI-AR | 0.05 | 4.14 | | 6.5 | 34.71 |
| PC-33:2 | 0.04 | 3.71 | | 5.83 | 40.54 |
| MGDG-28:0 | 0.04 | 3.44 | | 5.41 | 45.95 |
| PE-DAG-33:2 | 0.03 | 2.52 | | 3.97 | 49.92 |
| PE-DAG-31:2 | 0.03 | 2.14 | | 3.37 | 53.28 |

### Groups Cluster 1 & Cluster 2 — Average dissimilarity = 59.17

| IPLs | Average Cluster 1 | Average Cluster 2 | Average Dissimilarity | Dissimilarity/SD | Contribution (%) | Cumulative (%) |
|---|---|---|---|---|---|---|
| PC-35:0 | 0.02 | 0.08 | 3.18 | 1.34 | 5.37 | 5.37 |
| PE-DAG-32:1 | 0.06 | 0.02 | 2.35 | 1.73 | 3.98 | 9.35 |
| PI-AR | 0.06 | 0.02 | 2.21 | 1.74 | 3.73 | 13.08 |
| PG-DAG-36:2 | 0.05 | 0.02 | 1.98 | 1.64 | 3.35 | 16.43 |
| DGCC-27:0 | 0.04 | 0 | 1.93 | 1 | 3.26 | 19.69 |
| PC-36:2 | 0.01 | 0.05 | 1.79 | 1.57 | 3.02 | 22.71 |
| PC-34:1 | 0 | 0.04 | 1.79 | 1.03 | 3.02 | 25.73 |
| PC-32:1 | 0.01 | 0.03 | 1.36 | 5.58 | 2.3 | 28.03 |
| DGCC-26:0 | 0.04 | 0.01 | 1.34 | 0.95 | 2.27 | 30.3 |
| PC-35:1 | 0.01 | 0.03 | 1.27 | 0.9 | 2.15 | 32.45 |
| PE-DAG-33:1 | 0.03 | 0.01 | 1.02 | 1.2 | 1.73 | 34.18 |
| PC-33:1 | 0 | 0.02 | 0.96 | 7.61 | 1.63 | 35.8 |
| DGCC-28:0 | 0.02 | 0 | 0.93 | 1.28 | 1.57 | 37.37 |
| PC-AEG-34:3 | 0.02 | 0 | 0.9 | 1.03 | 1.52 | 38.89 |
| PE-DAG-34:2 | 0.03 | 0.02 | 0.88 | 1.2 | 1.49 | 40.38 |
| MGDG-32:1 | 0 | 0.02 | 0.83 | 1.81 | 1.4 | 41.78 |
| PC-30:1 | 0.03 | 0.01 | 0.83 | 1.15 | 1.4 | 43.18 |
| PG-DAG-34:2 | 0.02 | 0 | 0.77 | 1.05 | 1.3 | 44.48 |
| PE-DAG-33:0 | 0.02 | 0 | 0.76 | 1.11 | 1.29 | 45.77 |
| PG-DAG-35:1 | 0.02 | 0.01 | 0.74 | 1.22 | 1.26 | 47.03 |
| PE-DAG-34:1 | 0.02 | 0 | 0.74 | 2.06 | 1.25 | 48.27 |
| PC-26:0 | 0.02 | 0 | 0.72 | 1.74 | 1.21 | 49.48 |
| DGCC-30:0 | 0 | 0.01 | 0.68 | 1.32 | 1.15 | 50.64 |

### Groups Cluster 1 & Cluster 3 — Average dissimilarity = 60.69

| IPLs | Average Cluster 1 | Average Cluster 3 | Average Dissimilarity | Dissimilarity/SD | Contribution (%) | Cumulative (%) |
|---|---|---|---|---|---|---|
| DGCC-42:6 | 0 | 0.16 | 8.02 | 3.2 | 13.21 | 13.21 |
| PC-35:0 | 0.02 | 0.08 | 3.05 | 1.87 | 5.02 | 18.23 |
| PI-AR | 0.06 | 0.05 | 2.66 | 1.6 | 4.39 | 22.62 |
| PE-DAG-32:1 | 0.06 | 0.01 | 2.49 | 1.74 | 4.1 | 26.72 |
| PG-DAG-36:2 | 0.05 | 0.02 | 1.9 | 1.49 | 3.14 | 29.86 |
| DGCC-27:0 | 0.04 | 0.01 | 1.84 | 0.97 | 3.03 | 32.89 |
| DGCC-26:0 | 0.04 | 0.01 | 1.59 | 1.12 | 2.62 | 35.52 |
| PC-33:2 | 0 | 0.03 | 1.58 | 1.7 | 2.61 | 38.12 |
| PE-DAG-34:2 | 0.03 | 0.01 | 1.13 | 1.35 | 1.86 | 39.98 |
| PE-DAG-33:1 | 0.03 | 0.01 | 1.07 | 1.33 | 1.76 | 41.75 |
| PC-AEG-34:3 | 0.02 | 0 | 0.95 | 1.08 | 1.57 | 43.31 |
| PC-29:2 | 0.02 | 0.03 | 0.95 | 1.88 | 1.56 | 44.87 |
| PC-28:0 | 0.02 | 0 | 0.9 | 1.25 | 1.49 | 46.36 |
| PC-30:1 | 0.03 | 0.03 | 0.87 | 1.35 | 1.43 | 47.79 |
| PE-DAG-33:0 | 0.02 | 0 | 0.76 | 1.07 | 1.26 | 49.05 |
| PG-DAG-34:2 | 0.02 | 0.01 | 0.76 | 1.1 | 1.26 | 50.3 |

### Groups Cluster 1 & Cluster 4 — Average dissimilarity = 62.47

| IPLs | Average Cluster 1 | Average Cluster 4 | Average Dissimilarity | Dissimilarity/SD | Contribution (%) | Cumulative (%) |
|---|---|---|---|---|---|---|
| DGCC-42:6 | 0 | 0.14 | 6.99 | 2.57 | 11.19 | 11.19 |
| PC-30:2 | 0.01 | 0.12 | 5.66 | 3.64 | 9.06 | 20.24 |
| PE-DAG-32:1 | 0.06 | 0 | 3.17 | 2.09 | 5.07 | 25.31 |
| PC-35:0 | 0.02 | 0.04 | 2.22 | 1.6 | 3.55 | 28.86 |
| PG-DAG-36:2 | 0.05 | 0.01 | 2.12 | 1.64 | 3.4 | 32.27 |
| PC-33:2 | 0 | 0.04 | 1.9 | 15.16 | 3.04 | 35.3 |
| DGCC-27:0 | 0.04 | 0.02 | 1.45 | 0.78 | 2.32 | 37.62 |
| PE-DAG-34:2 | 0.03 | 0 | 1.35 | 1.44 | 2.16 | 39.78 |
| PI-AR | 0.06 | 0.05 | 1.3 | 1.6 | 2.08 | 41.86 |
| DGCC-26:0 | 0.04 | 0.01 | 1.26 | 0.89 | 2.02 | 43.88 |
| DGDG-34:2 | 0 | 0.03 | 1.25 | 1.17 | 2 | 45.88 |
| PE-DAG-31:2 | 0 | 0.03 | 1.21 | 4.58 | 1.93 | 47.81 |
| PE-DAG-33:1 | 0.03 | 0.01 | 1.2 | 1.46 | 1.92 | 49.73 |
| PE-DAG-33:3 | 0 | 0.02 | 1.16 | 4.61 | 1.86 | 51.59 |





### 3.3 Distribution of alkyl chains based on length and degree of unsaturation

The difference in the total number of acyl carbon atoms in both alkyl chains, rather than in individual fatty acids, and in the number of acyl double bonds within each cluster is shown in Fig. 4. Statistical differences of IPLs classes within each cluster was obtained through a Tukey HSD post-hoc test at a significant level of $p < 0.05$ (Fig. 4a, b). The average number of carbon atoms in the diglyceride moieties of IPLs in the Cluster 1 showed that DGCC, MGDG, Others, PC, and PG were all distinct from one another ($n = 283$; $P < 0.05$; Fig. 4a). PG and Others were characterized by relatively long alkyl chains (35-36 C atoms, respectively) and DGCC for shorter alkyl chains (32 C atoms). In general, Cluster 1 exhibited a wide range of chain lengths among diacylglycerols (DAGs) (28-36 C atoms). Cluster 2 showed narrower ranges than Cluster 1 (30-36 C atoms). This cluster also showed no statistical difference ($p > 0.05$) among IPL classes (Fig. 4a), following pairwise comparisons with Tukey's HSD post-hoc test, despite the wide range of DGCC structures. Cluster 3, while it exhibited low variability in betaine lipids, showed the highest number of carbon atoms in DGCCs (42). On the contrary, Cluster 4 showed high viability in DGCCs, which did not exceed 42 carbon atoms. Within the phospholipid class, PG showed the highest number of carbon atoms in all clusters, the mean we observed was 34 carbon atoms and a range of 32 to 37 (Fig. 4a). The un-cluster sample (hadal sediment of 0-1 cm at A4 station) was characterized by relatively longer alkyl chains (up to 42 C atoms) than Cluster 1 (Fig. 4a).

Overall, the degree of unsaturation (i.e., number of double bounds) within clusters was variable (Fig. 4b). Cluster 1 predominantly consisted of fully saturated and mono-unsaturated IPLs, except for PG that showed 2 double bonds in average. In Cluster 2, the fatty acids of DGCCs were distinctly variable, although they exhibited 2 unsaturations on average. A similar pattern was observed in DGDGs with an average of 2.5 unsaturations (Fig. 4b). DGTS, MGDGD, PC and SQ showed non to 1 unsaturation, whereas DGTA, PE and PG exhibited between 1 and 2.5 unsaturations. Cluster 3 showed more than 5 unsaturations on average for DGCC, unlike other IPL classes that did not exceed 2 unsaturations. In Cluster 4, PG and DGCC presented ~3 and ~5 unsaturations on average. Also, on average, DGDG and SQ exhibited 2 unsaturations, MGDG and Others were mono-unsaturated, and DGTS were saturated (Fig. 4b). Additionally, the ratio of total unsaturated fatty acids to total saturated fatty acids in IPLs increased from (on average) ~0.9 in all water column samples (pressure range of 2-76 Bars) to ~2.7 in the bathyal (pressure range of 54-113 Bars) and hadal sediments (pressure range 777-810 Bars) (Fig. 5).



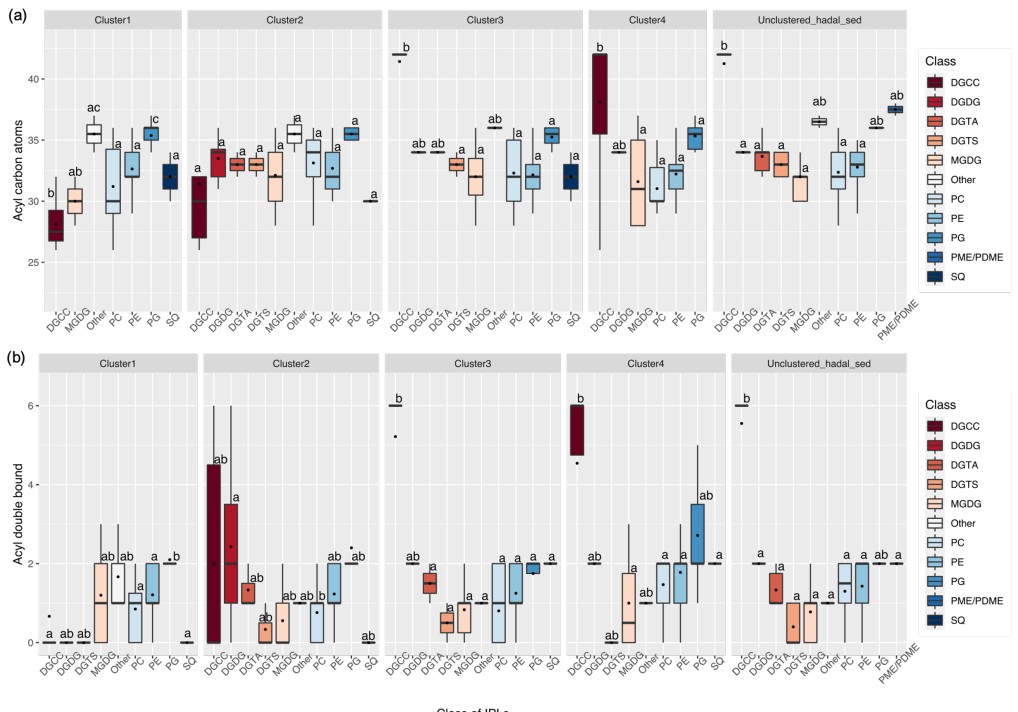

**Figure 4. Total number of acyl carbon atoms (a) and acyl double bonds (b) in IPL classes across the distinct clusters**
**shown in Fig. 2. The letters "a" and "b" in the plots indicate the presence of statistically distinct groups (p < 0.05) from**
**both ANOVA and post-hoc Tukey HSD tests, respectively.**



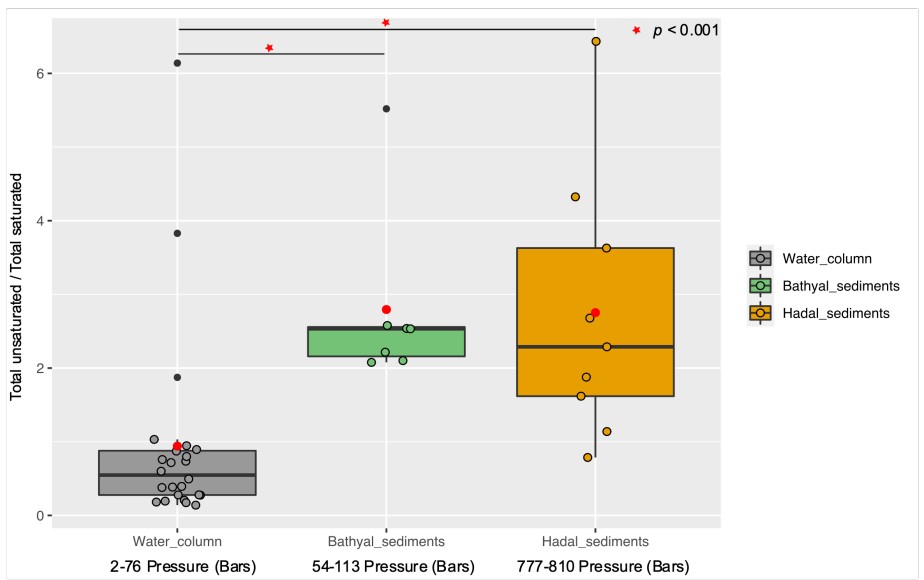

**Figure 5. Boxplot showing difference in ratio of total unsaturated fatty acids to total saturated fatty acids derived from IPL across water column (Cantarero et al., 2020) and sediments of Atacama Trench (this study). The average in each environment were shown in red circles and Wilcoxon test (p-value < 0.001) shows that the sediments have statistical ratios higher than the water column.**

### 3.4 Unique IPLs in hadal sediments of the Atacama Trench

To test whether IPLs detected in the Atacama Trench (AT) exhibit signals unique to this environment, we compared our results against IPL data from the overlying water column (top 750 meters) recently reported by Cantarero et al. (2020). Water-column particles and bathyal-hadal sediments shared 242 (96.1%) IPL structures (Fig. 6a), while hadal sediments and water-column particles shared 14 (0.02%), and hadal and bathyal sediments shared 55 (3.6%). Of all the analyzed IPLs reported in this study, eight of them were unique to the Atacama Trench sediments and not present in shallower sediments nor the overlying water column. They include five glycolipids (SQDG-42:11, SQDG-23:0, DGDG-35:1, DGDG-35:2 and DGDG-37:1), two phosphatidyl-inositols (PI-diOH-Ext-AR and PI-OH-AR), and one ornithine lipid (OL-37:6). While unique to hadal sediments, their total concentration was low (~53.32 ng g$^{-1}$ sediment) and they contributed ~0.00012% of the total IPL pool (Fig. 6a). We then performed a cluster analysis to compare IPLs in deep-sea surface sediments against IPLs reported in the overlying water column (Cantarero et al., 2020; Fig. 6b). Cluster 1 comprised samples from the core OMZ in the free-living fraction (AU p-value of 100%). Cluster 2 comprised samples from both the upper and lower oxyclines (~14-60 m) as well as from the chlorophyll maximum (AU p-value of 99%). Cluster 3 comprised bathyal and hadal samples (AU p-value of 99%). Cluster 4 mostly comprised the deepest water column sample (mesopelagic region at 750 m) and hadal samples (AU p-value of 98%; Fig. 6b). We also compared IPLs in hadal and bathyal sediments against the pool of IPLs reported as diagnostic of the planktonic community inhabiting the chlorophyll maximum in the upper water column (Cantarero et al., 2020), and thus assess their export and stability through





their transit to the deep-sea. Notably, these IPLs only represent ~0.001-0.005% and 0.002-0.03% of the total IPL
pool in hadal and bathyal sediments, respectively (Fig. S3).

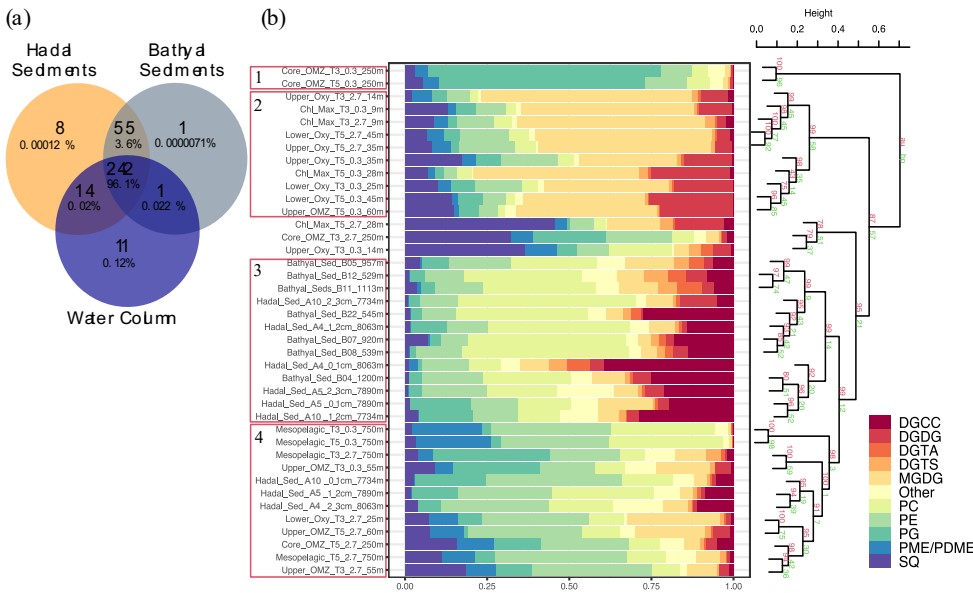


**Figure 6. Comparison of IPLs in sediments (this study) and the overlying water column (Cantarero et al., 2020). (a)**
**Venn diagram showing unique and shared IPL molecules in hadal and bathyal sediments and the water column. (b)**
**Cumulative bar charts of IPL fractional abundances in each sampling station. Samples were grouped according to**
**arithmetic mean (UPGMA) hierarchical clustering based on Euclidean distances. The cluster analysis on the right-**
**hand size shows approximately unbiased (AU) and bootstrap probability (BP) in red and green numbers, respectively,**
**as well as p-values are shown at branching points. Clusters with AU ≥ 95% confidence are highlighted in red on the**
**left-hand side.**
In sediments within the AT, we found a high degree of heterogeneity in total IPL concentrations among sites and
different sediment levels (0–1, 1–2, 2–3 cm), which were an order of magnitude higher than bathyal sediments
(see Figs. S4a, S4b). Hadal sediments at station A10 (7,734 m) showed a large range of phospholipid
concentrations (~47–2,698 ng g$^{-1}$ sediment) (supplementary Fig. S4b). Although the highest total IPL abundances
were observed at hadal station A10 (Fig. S4b), the greatest diversity in IPL composition was observed in the 0-1
cm of the hadal station A4, previously referred to as un-clustered (see Fig. 2). The most abundant IPL class in
hadal sediments were phospholipids, PCs (~41-2,698 ng g$^{-1}$ sed.), PEs (~26-1,813 ng g$^{-1}$ sed.) and PGs (5-937 ng
g$^{-1}$ sed). The concentration of IPLs normalized by TOC (ng IPL/g TOC) showed maximum values in the hadal
station A10 (~150 ng IPL/g TOC), followed by lower values in the hadal stations A5 and A4 of ~50 and ~12 ng
IPL/g TOC, respectively (Fig. S5).





**4. Discussion**

**4.1 Potential Biological Sources of IPLs**

We evaluated the chemical characteristics and potential biological provenance of labile IPLs in the hadal zone of
the AT as compared to the overlying water column and/or shallower bathyal sediments.

**4.2 Potential sources of phospholipids**

**PG (Phosphatidylglycerol)**

Phospholipids are common constituents of cellular membranes in most microorganisms (Ratledge and Wilkinson,
1988). Since PGs play an essential role in photosynthesis (Wada and Murata, 2007), they have therefore been
mainly identified in algal and bacterial photoautotrophs (Dowhan, 1997; Sato et al., 2000; Gombos et al., 2002).
However, their biological origin is highly diverse and also includes heterotrophic bacteria (Oliver and Colwell,
1973; Van Mooy et al., 2009; Popendorf et al., 2011b; Carini et al., 2015; Sebastián et al., 2016), methylotrophs
(Batrakov and Nikitin, 1996), methanotrophic bacteria (Makula, 1978), *Pelagibacter ubique* (Van Mooy et al.,
2009), and barophilic bacteria (e.g., DB21MT-2 and DB21MT-5) isolated from sediments from the Marianas
Trench (Fang et al., 2000).

The hierarchical cluster analysis on variations in the relative abundance of PGs suggests that several compounds
maintained a similar proportion in bathyal and hadal sediments, which differs from the water column (Fig. 7a).
Most PGs in the bathyal and hadal sediments have long acyl carbon chains ($C_{34}$-$C_{41}$), and they show odd- and
even-numbered polyunsaturated fatty acids (Fig. 7a). The average chain-lengths of even-numbered $n$-$C_{18}$, $n$-$C_{20}$
and $n$-$C_{22}$ fatty acids, mostly in PCs and PGs, are indicative of algal inputs (Kaneda, 1991; Thompson Jr, 1996;
Bergé and Barnathan, 2005; Brandsma et al., 2012). However, since these PGs were not dominant in the water
column, a source from deeper environments is likely. Specifically, PG-DAG-36:2, PG-DAG-35:2, PG-DAG-36:5,
PG-DAG-37:2, and PG-DAG-41:4 are the dominant constituents of this IPL class in hadal-bathyal sediments (Fig.
7a). PG-DAG-36:2 has been described in surface waters of the North Sea and also detected in picoeukaryotes
(Brandsma et al., 2012), and in heterotrophic bacteria in surface waters of the open South Pacific Ocean (Van
Mooy and Fredricks, 2010). However, these PGs are not dominant in the water column near the Atacama Trench
(Cantarero et al., 2020). On the other hand, PG-DAG-35:2, PG-DAG-36:5, PG-DAG-37:2 and PG-DAG-41:4 are
not commonly reported in water-column studies. Thus, it is possible that PGs present in the AT sediments derive
from *in situ* microbial production, and/or from the transport of labile organic matter present in surface sediment
at shallower depths. PG-DAG-36:2 (Fig. 3) is the PG contributing most to the dissimilarity within the cluster
containing only hadal sediments (Cluster 1 in Figure 2). Thus, this lipid appears to be more representative of *in*
*situ* microbial production in this environment.

**PE (Phosphatidylethanolamine)**

PE and its methylated derivatives (PME, PDME) have been predominantly reported in membranes of diverse
bacterial sources, including heterotrophic bacteria (Van Mooy and Fredricks, 2010; Schubotz et al., 2018a),
nitrifying/denitrifying bacteria (Goldfine and Hagen, 1968), sulfate-reducing bacteria (Rütters et al., 2001; Sturt



et al., 2004), sulfur-oxidizing bacteria (Barridge and Shively, 1968; Imhoff, 1995; Wakeham et al., 2012),
methanotrophic bacteria (Makula, 1978; Sturt et al., 2004), and barophilic bacteria (Fang et al., 2000).

PEs showed a similar distribution in bathyal and hadal sediments (Fig. 7b), where they are dominated by long-
chain ($C_{30-40}$) polyunsaturated fatty acids, contrary to the shorter chains ($C_{29-32}$) of saturated and monounsaturated
fatty acids present in the water column. PE-DAG-32:1, PE-DAG-32:2, and PE-DAG-33:1 are the dominant PE
compounds of bathyal and hadal sediments. These IPLs have been previously reported in heterotrophic bacteria
(Van Mooy and Fredricks, 2010; Brandsma et al., 2012). On the other hand, fatty acids in PEs including
monounsaturated and polyunsaturated (e.g., $C_{20:5}$ and $C_{22:6}$) have been reported in barophilic bacteria isolated from
sediments from the Marianas Trench (Fang et al., 2000). Thus, although we cannot confidentially rule out other
sources, it is possible that PEs present in the AT sediments derive from *in situ* production by barophilic
heterotrophic bacteria, and/or from the transport of labile organic matter from shallower water sediment. PE-
DAG-32:1, PE-DAG-32:2 and PE-DAG-33:1 (Fig. 3) are the PEs that contributed most to the dissimilarity within
the cluster containing only hadal sediment samples (Cluster 1 in Figure 2). Thus, this cluster appears to be
representative of *in situ* microbial production in this environment.

**PC (Phosphatidylcholine)**

PCs were amongst the most diverse (43 structures: Fig. 7c) and abundant phospholipid class in hadal sediments
(Fig. S4). PC is the major membrane-forming phospholipid in eukaryotes (Lechevalier, 1988; Sohlenkamp et al.,
2003; Van Mooy et al., 2006; Van Mooy and Fredricks, 2010). Additionally, PC has been reported to be a major
DAG in zooplankton, from protozoa to copepods and krill (Patton et al., 1972; Mayzaud et al., 1999; Lund and
Chu, 2002). However, genomic data indicates that more than 10% of all bacteria possess the genetic machinery
for PC biosynthesis (Sohlenkamp et al., 2003). PC has also been reported in nitrifying bacteria (Lam et al., 2007),
photoheterotrophic bacteria (Koblížek et al., 2006; Van Mooy et al., 2006), and barophilic bacteria (Fang et al.,
2000). In surface sediments of the Black Sea (2000 m), PCs were related to algal material rapidly exported from
surface waters (Schubotz et al., 2009a).

Hadal and bathyal sediments, in addition to two OMZ core stations, were clustered in the PC class (AU p-value
of 97%; Fig. 7c). This cluster showed PCs with long ($C_{33-38}$) and polyunsaturated fatty acids (up to 10
unsaturations). The dominant constituents were PC-35:0, PC-30:2, PC-30:1, PC-33:2, PC-35:1, PC-29:2, PC-32:1,
and PC-36:2 (Fig. 7c). PC-36:2 and PC-30:1 has been associated with phytoplankton detritus (Schubotz et al.,
2009a) and bacteria (Brandsma et al., 2012), whereas PC-32:1 has been associated with picoeukaryotes (Brandsma
et al., 2012). Since the most abundant PCs in Cluster 1 have not been reported as dominant structures in any
specific environment before, they are possibly produced *in situ* and/or derive from laterally transported sediment
from shallower depths to the hadal region. Among bacteria, those membranes reported to contain PC belong to
the alpha and gamma subgroups of the proteobacteria (Sohlenkamp et al., 2003). Given that these bacterial groups
are abundant in trench samples from Puerto Rico (Eloe et al., 2011), the Mariana (Nunoura et al., 2015) and
recently in the Atacama Trench (Schauberger et al., 2021), it is possible that PCs present in high abundance in the
AT trenches is consistent with high abundance of proteobacteria in these regions. Given their general known

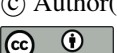



association and abundance in AT sediments (Fig. S4), they likely derive primarily from bacterial sources and to a
lesser extent from phytoplankton.

**PME/PDME (Phosphatidyl(di)methylethanolamine)**

PME/PDMEs have been observed in association with methanotrophic bacteria (Makula, 1978; Goldfine, 1984;
Fang et al., 2000), sulfide oxidizer bacteria (Barridge and Shively, 1968), sulfate-reducing bacteria, mainly
*Desulfobulbus spp* (Rossel et al., 2011), Proteobacteria (Oliver and Colwell, 1973; Goldfine, 1984), and barophilic
bacteria from the Mariana Trench (Fang et al., 2000). Additionally, the occurrence of PME-DEG at some hadal
stations suggests the presence of sulfate-reducing bacteria (Rütters et al., 2001; Sturt et al., 2004).
PME/PDMEs exhibited their lowest abundance ($\sim$10 ug g sed$^{-1}$) in sediment samples (Fig. S4b). In the bathyal
and hadal sediments they were clustered (AU p-value of 97%) and dominated by PDME-DAG-33:1, PME-DAG-
37:2, PME-DAG-34:2, PME-DAG-31:1, and PME-DEG-33:0 (Fig. S6a). PME-DEG-33:0 has been shown to
correlate with high $NO_2^-$ in the overlying water column of this area (Cantarero et al., 2020), which could suggest
a potential association with denitrification processes. These structures have also been reported in the deep
chemocline of the Cariaco basin (Wakeham et al., 2012), suggesting a potential chemoautotrophic and/or
heterotrophic source. These signals are different from the water column, which is dominated by the saturated
PME-32:0, PME-DAG-30:0, and PME-DAG-31:0 (Fig. S6a and S7; Cantarero et al., 2020). Thus, they most
likely derive from *in situ* production and or/lateral transport rather than vertical export from the overlying water
column. No particular PME/PDME were found to contribute to the dissimilarity between the cluster containing
only hadal sediment samples (Cluster 1 in Figure 2) and other sediment samples.




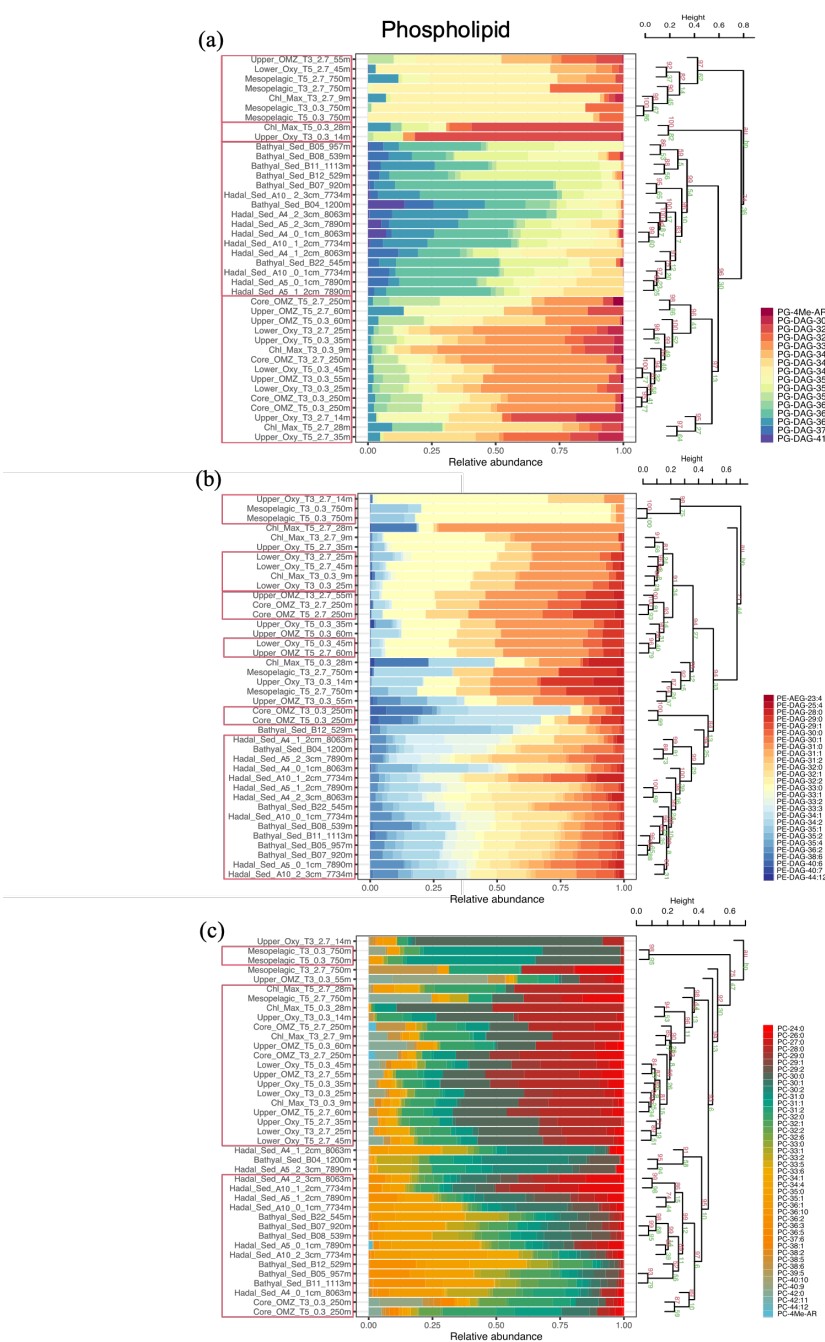


Figure 7. Cumulative bar chart of phospholipid fractional abundances in all sample types. a) PG; b) PE; c) PC. The
number of carbon atoms and unsaturation in core fatty acids follow the order shown in the legend. The right panel
depicts a cluster analysis with approximately unbiased (AU) and bootstrap probability (BP) shown in red and green,
respectively, and p-values shown at branching points. The number of bootstrap replicates is 10000. Clusters with AU
≥ 95% confidence are highlighted in red boxes on the left-hand side.



### 4.3 Potential sources of glycolipids

**MGDG (Monoglycosyldiacylglycerol)**

Due to their dominant occurrence in chloroplast thylakoid membranes (Murata and Siegenthaler, 1998) and particularly in cyanobacteria (Heinz, 1977; Harwood, 1998; Wada and Murata, 2007; Van Mooy and Fredricks, 2010), but also in heterotrophic bacteria (Popendorf et al., 2011b), MGDGs are probably the most abundant IPLs on earth (Gounaris and Barber, 1983).

The hierarchical cluster groups of MGDGs in bathyal (AU p-value of 90%) and hadal (AU p-value of 98%) sediments (Fig. 8a). The most abundant MGDGs in the bathyal and hadal sediments were MGDG-28:0, MGDG-32:1, MGDG-30:1, MGDG-32:0 and MGDG-37:3. MGDG-28:0, and MGDG-30:1 are ubiquitous along the oxycline of the overlying OMZ (Cantarero et al., 2020) and MGDG-32:1. MGDG-32:0 have been reported in waters of the eastern south Pacific (Van Mooy and Fredricks, 2010). Thus, these MGDGs could indicate the occurrence of at least some export of labile organic matter from surface waters in our study area. MGDG-37:3 does not appear to be a dominant structure in any specific environment previously reported, possibly being produced *in situ* and /or laterally transported to hadal sediments.

**DGDG (Diglycosyldiacylglycerol)**

DGDGs are commonly found in membranes of eukaryotic algae and cyanobacteria (Wada and Murata, 1998; Sakurai et al., 2006; Kalisch et al., 2016). DGDGs clustered together in bathyal and hadal sediments (AU p value of 96%) whereas their distribution differed from the water column (Fig. 8b).

The most abundant DGDGs in hadal and bathyal sediments of the AT were DGDG-34:2, which has been previously reported in Chlorophyta (da Costa et al., 2020), and DGDG-30:0, which is widely distributed in the water column of this region (Cantarero et al., 2020). Thus, although they account for less than 10% of the total IPL pool (Fig. 6b), their presence in bathyal and hadal sediments is likely indicating the occurrence of at least some export of labile organic matter from surface waters.

**SQDG (Sulfoquinovosyldiacylglycerol)**

SQDG are predominantly produced by photoautotrophs (Van Mooy et al., 2006; Popendorf et al., 2011b), including various groups of diatoms, brown and green algal chloroplast membranes (Harwood, 1998), and cyanobacteria (Siegenthaler, 1998; Wada and Murata, 1998). SQDGs have also been found in bacteria from the α- and γ-proteobacterial lineages (Benning, 1998). In the overlying water column of the Atacama Trench, Cantarero et al., (2020) suggested a higher contribution of SQDGs from cyanobacteria than algae. Also, SQDGs found in the deep Atlantic (down to ~4,000-5,000 m) appear to indicate a source and export from surface waters (Gašparović et al., 2018).

SQDGs showed a consistent distribution in bathyal and hadal sediments, where they are dominated by long-chain ($C_{30-42}$) fatty acids (Fig. 8c). This is contrasting to their distribution in the overlying water column where they are dominated by shorter chains ($C_{28-32}$) of saturated fatty acids (Cantarero et al., 2020). SQDG-30:0, SQDG-32:0, SQDG-30:2, and SQDG-38:4 were the dominant SQDG constituents of bathyal and hadal sediments. SQDG-30:0 and SQDG-30:2 have been reported in bacteria in North Sea surface waters (Brandsma et al., 2012), in cyanobacteria of the eastern subtropical South Pacific (Van Mooy and Fredricks, 2010), and in plankton detritus



from surface sediments of the Black Sea (Schubotz et al., 2009a). Furthermore, SQDG-30:0 is abundant in surface
waters of our study area and SQDG-38:4 has been correlated with $NO_3^-$ (Cantarero et al., 2020).
The observed differences in the distribution of SQDGs in deep sediments compared to the water column suggests
an *in situ* production of previously poorly characterized compounds, and/or lateral transport in addition to at least
some export from surface waters.





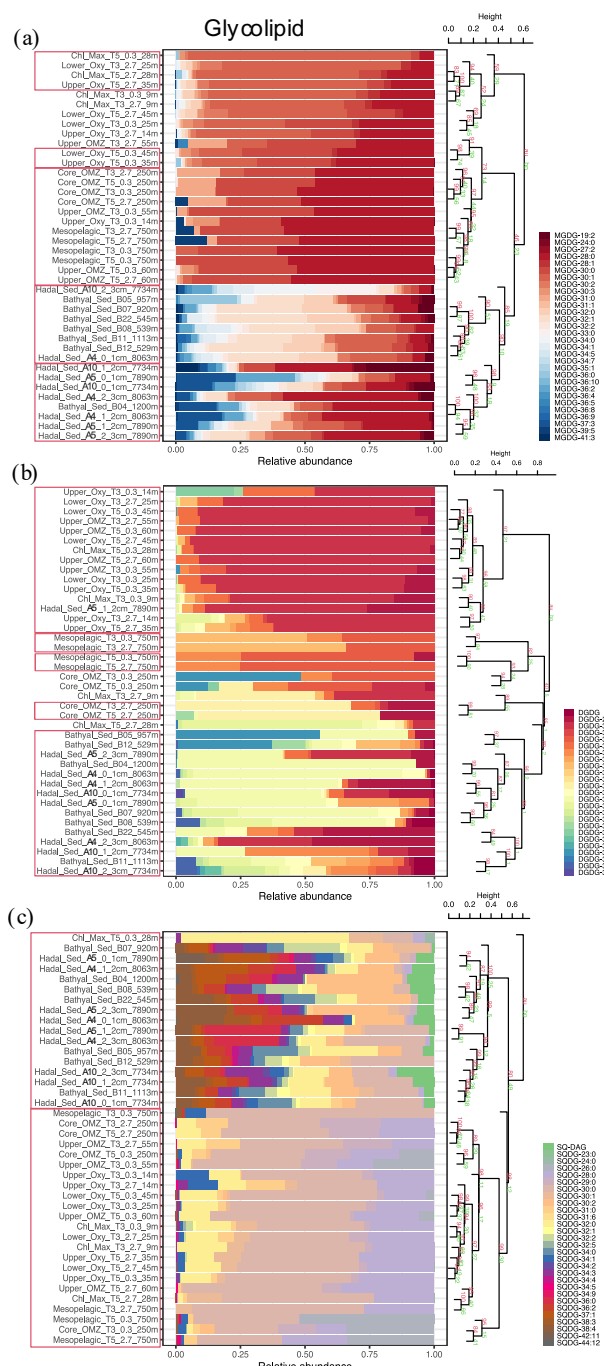

**Figure 8.** Cumulative bar chart of glycolipid fractional abundances at each sampling station. A) MGDG; B) DGDG; C) SQDG. The number of carbon atoms and unsaturations in core fatty acids follows the order shown in the legend. The right panel depicts a cluster analysis with approximately unbiased (AU) and bootstrap probability (BP) in red and green, respectively, and p-values shown at branching points. The number of bootstrap replicates is 10000. Clusters with AU ≥ 95% confidence are highlighted in red boxes on the left-hand side.

**4.4 Potential biological sources of betaine lipids**



**DGTS (Diacylglyceryl trimethylhomoserine)**

DGTSs have diverse biological origins, being found in a wide range of eukaryotes (Sato, 1992; Dembitsky, 1996; Kato et al., 1997; Van Mooy et al., 2009), photoheterotrophic bacteria (Benning et al., 1995; Geiger et al., 1999), photoautotrophic bacteria (Popendorf et al., 2011b) including cyanobacteria (Řezanka et al., 2003), and members of the α-Proteobacteria subdivision (López-Lara et al., 2003). Schubotz et al. (2018) showed DGTS with varying fatty-acid compositions in the OMZ system of the eastern tropical North Pacific, especially in OMZ waters, indicating that these compounds can be biosynthesized by a wider range of source organisms than previously thought.

Consistent with other IPL classes, DGTSs of the bathyal and hadal samples were grouped in the same cluster (AU p-value of 98%) and differed from the water column (Fig. 9a). However, several DGTSs are shared between surface waters (9-60 m) and deep sediments. Indeed, the most abundant DGTSs in bathyal and hadal sediments (DGTS-34:0, DGTS-32:1, DGTS-26:0, DGTS-34:1, DGTS-32:0, and DGTS-25:0; Fig. 9a) are also prominent in the chlorophyll maximum in the eastern subtropical South Pacific (Van Mooy and Fredricks, 2010, and Cantarero et al., 2020). Therefore, their presence in hadal sediments suggest the export of labile organic matter from the euphotic zone, although we cannot rule out other sources.

**DGTA (Diacylglyceryl hydroxymethyl-trimethyl-β-alanine)**

DGTAs have been widely reported in eukaryotic phytoplankton (Araki et al., 1991; Dembitsky, 1996; Cañavate et al., 2017), mainly in diatoms (Volkman et al., 1989; Zhukova, 2005; Gómez-Consarnau et al., 2007), and are also especially abundant in cultures of Prymnesiophytes and Cryptophytes (Kato et al., 1997). DGTAs have also been found in cyanobacteria (Brandsma et al., 2012) and heterotrophic bacteria (Popendorf et al., 2011a; Sebastián et al., 2016).

DGTAs in bathyal and hadal sediments are mainly composed of longer ($C_{28}$-$C_{42}$) and polyunsaturated (1-12) fatty acids compared to those present in the shallowest region of the overlying water column, composed of shorter and saturated fatty acids (Fig. 9b). In the overlying water column, these compounds are associated with relatively high chlorophyll and $O_2$ concentrations (Cantarero et al., 2020), similar to North Sea surface waters (Brandsma et al., 2012). To the best of our knowledge, the dominant DGTAs in hadal and bathyal sediments (Fig. 9b) have not been previously reported as dominant IPLs in other environments. Whereas no specific biological sources in hadal sediments are known, the structures containing between 30 and 38 carbon atoms might be characteristic of this type of environment.

**DGCC (Diacylglycerylcarboxy-N-hydroxymethyl-choline)**

Our knowledge of DGCC sources is limited. They have been found in membranes of Prymnesiophyte algae (Kato et al., 1994), mainly in *Pavlova lutheria* (Kato et al., 1994; Eichenberger and Gribi, 1997), and in *E. huxleyi* (Volkman et al., 1989; Pond and Harris, 1996; Van Mooy and Fredricks, 2010). Additionally, they have also been reported in the diatom *Thalassiosira pseudonana* (Van Mooy et al., 2009).

The most abundant IPL from the entire data set of Bathyal and hadal sediments is DGCC-42:6 (Fig. 9c). This is the compound with more C atoms in all IPLs in this study, and with 6 unsaturations. DGCCs with long-chain, polyunsaturated fatty acids (i.e., $C_{38:6}$, $C_{40:10}$, $C_{42:11}$, $C_{44:12}$) have been previously reported in phytoplankton





(Hunter, 2015; Van Mooy and Fredricks, 2010). However, the most abundant DGCCs in hadal sediments have,
to the best of our knowledge, not been previously reported, which highlights their potential as biomarkers of deep-
sea sediments. However, 3 hadal stations clustered in a separate group (see Fig. 9c), were dominated by DGCC-
27:0, and did not contain DGCC-42:6, indicating that this IPL probably comes from lateral transport of organic
matter in the rest of the hadal samples.

**4.5 Potential biological sources of other lipids**

Glycosidic ceramides (Gly-Cer) have been reported in eukaryotic algae such as Prymnesiophyte (Vardi et al.,
2009), and have also been shown to be abundant in water columns of OMZ systems (Schubotz et al., 2009b, 2018;
Cantarero et al., 2020). In general, the overlying water column shows Gly-Cer with polyunsaturated fatty acids
with $C_{21-38}$. However, these structures are scarce in the bathyal and hadal sediments (see Fig. S6b), which could
reflect a deficient export from surface waters due to intense remineralization. On the other hand, Ornithine lipids
(OL), phosphatidylinositol (PI), PC-AEGs and other unidentified phospholipids were also present in deep
sediments (Fig. S6b). Some PIs and OLs have been reported in sulfate-reducing bacteria (Sturt et al., 2004;
Bühring et al., 2014), whereas PC-AEGs have been reported in bacteria inhabiting water columns with reduced
oxygen concentration (Schubotz et al., 2018b). Thus, the high relative abundance of PI-AR and PC-AEG-34:3 in
hadal and bathyal sediments (Figs. S6b and S7) could be indicative of anaerobic microbial processes. In particular,
PI-AR have been related to archaea (Morii et al., 2014), suggesting a high abundance of these microbes in hadal
and bathyal sediments. Members of the archaeal domain have already been reported to be abundant in hadal
sediments (Xu et al., 2020), however, GDGTs are better markers to explore the IPL content of their membranes
(Liu et al., 2011) than those we use here, which warrants further investigation.

Although we cannot confidentially rule out other sources, it is likely that these IPLs in the AT sediments are
derived from *in situ* microbial processes, and/or from the transport of labile organic matter from shallower
sediment. In particular PI-AR and PC-AEG-34:3 contributed the most to the dissimilarity between the cluster
containing only hadal sediment samples (Cluster 1 in Figs. 2, and 3), thus suggesting an *in situ* microbial
production. While IPLs can derive from multiple biological sources, we lack data on the largely uncharacterized
hadal endemic microbial community, and existing IPL data from the water column does not take into account
temporal variability of their biological sources, our study represents a step forward on the characterization of
labile sources of organic matter sustaining hadal ecosystems.

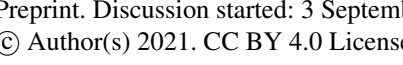



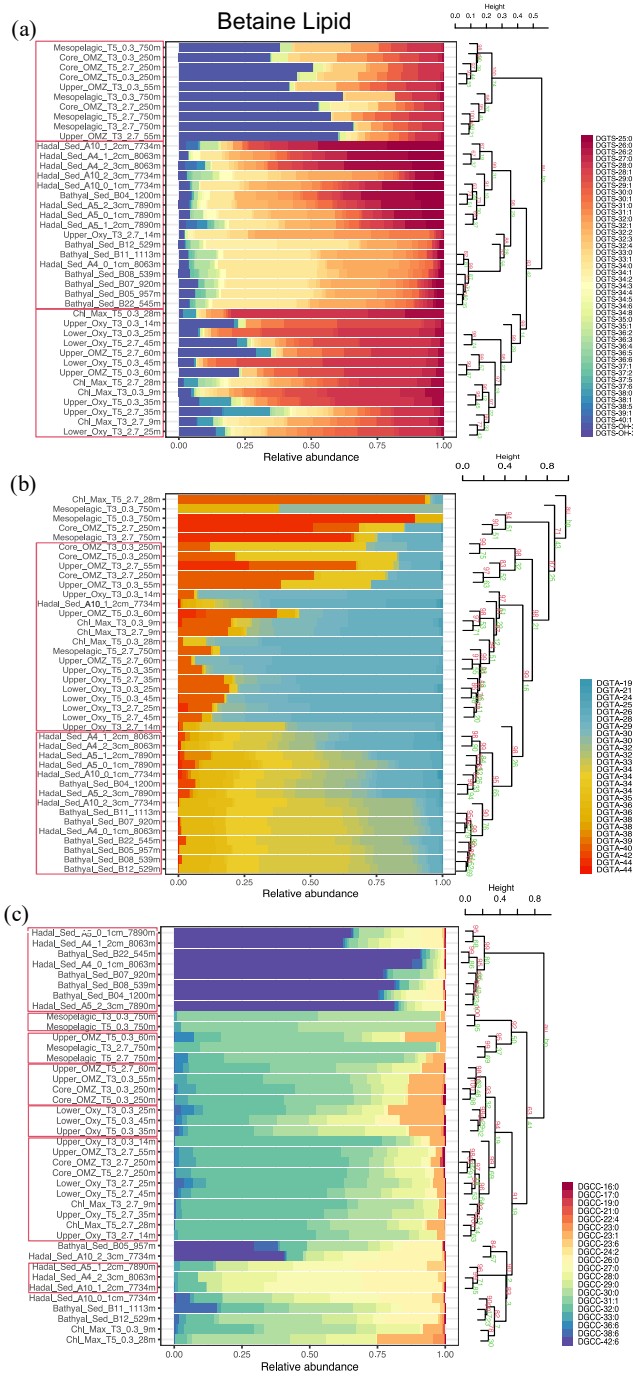


**Figure 9. Cumulative bar chart of betaine fractional abundances at each sampling station. A) DGTS; B) DGTA; C) DGCC. The number of carbon atoms and unsaturations core fatty acids follow the order shown in the legend. The right panel depicts a cluster analysis with approximately unbiased (AU) and bootstrap probability (BP) in red and green, respectively, and p-values shown at branching points. The number of bootstrap replicates is 10000. Clusters with AU ≥ 95% confidence are highlighted in red boxes on the left-hand side.**





### 4.6 Allochthonous versus autochthonous IPLs in the Atacama Trench

Given their rapid degradation after cell death (White et al., 1979; Harvey et al., 1986), IPLs are typically considered markers of living or recently dead cells (White et al., 1979; Harvey et al., 1986; Petersen et al., 1991; Lipp et al., 2008). The distribution of IPLs in bathyal and hadal sediments exhibits a high degree of similitude, as demonstrated by the hierarchical analysis (Cluster 1 in Fig. 10a), the NMDS (Fig. 10b), and the SIMPER analysis (Cluster 1 in Table S1). The deep-sea surface sediments showed weak clustering with the IPLs reported in the overlying water column by Cantarero et al. (2020) (Fig 9a). Additionally, water column samples exhibit a larger degree of separation than sediments (ANOSIM, R = 0.78; P <0.01; Fig. 10b) and are broadly clustered by geochemical environments (Cantarero et al., 2020). The low abundance of IPLs characteristic of organisms inhabiting the chlorophyll maximum in deep-sea sediments of the AT (<0.005 % of the total IPL pool; Fig. S3) suggests minimal export of labile organic compounds from the upper ocean, likely due to intense degradation through transit in the water column. Indeed, by using the experimentally calculated kinetic degradation rate constants ($k'$) of ester-bound IPLs by Logemann et al. (2011), and the sinking rate of particles from surface waters to 4000 m (20-100 m/day; Billett et al., 1983; Danovaro et al., 2014), we calculated that ~86-98% ($k'_{t=80}$= 0.033 and $k'_{t=400}$= 0.011) of IPLs from surface waters should degrade by the time that particles reach depths of ~8000 m. These results are in accord with studies indicating elevated benthic oxygen consumption rates resulting from intense microbial respiration of sinking organic matter reaching the sediment (Glud et al., 2013; Wenzhöfer et al., 2016). Thus, the pool of IPLs in hadal sediments appears to predominantly represents *in situ* microbial production and/or lateral sediment transport from shallower depths, and is likely that the deep-sea microbial communities in both benthic and hadal sediments is similar despite their bathymetric zonation (~1,000-8,000 m). Alternatively, we cannot rule out the possibility of new IPL production, particularly from heterotrophic and chemoautochthonic bacteria in micro niches of sinking particles, reaching the deep-sea.

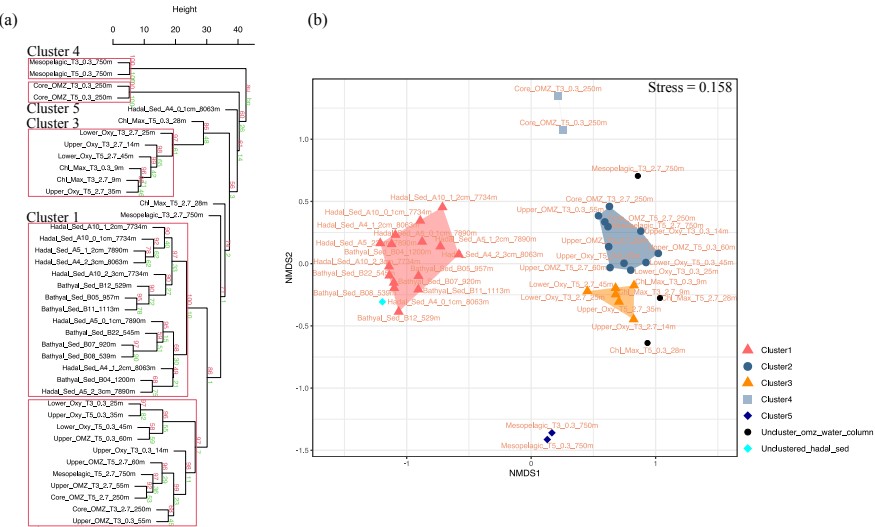

**Figure 10.** (a) Arithmetic mean (UPGMA) hierarchical clustering based on Euclidean distances calculated from IPLs in each sampling station. Red values are Approximately Unbiased (AU) p-values and green values are Bootstrap Probability (BP) for



each node. Red boxes highlight clusters with 95% confidence. The number of bootstrap replicates is 10000. (b) Non-metric
multidimensional scaling (NMDS) analysis of IPLs at each sampling station. The distance matrix was calculated based on the
Bray–Curtis dissimilarity. The stress value of the final configuration was 15.8 %. Different symbols and colors represent the
sample grouping from hierarchical clusters shown in panel a.

Marine trenches receive organic carbon from a variety of sources and transport mechanisms. These include
canyons and river systems that channel organic matter from land to coastal regions, aeolian transport, surface
water productivity, and *in situ* production, to name a few (Wenzhöfer et al., 2016; Tarn et al., 2016; Luo et al.,
2017; Xu et al., 2018; Guan et al., 2019; Xu et al., 2021). Carbon flux events can increase the delivery of particulate
carbon from surface waters to the seafloor (Poff et al., 2021), whereas river discharge and aeolian transport can
result in enhanced terrestrial carbon (Xu et al., 2021). Mass wasting events are also known to create dynamic
depositional conditions and strong spatial heterogeneity in organic matter distribution in marine trenches
(Schauberger et al., 2021; Xu et al., 2021). While marine organic carbon appears to dominate sediments from the
Japan (Schwestermann et al., 2021) Massau, and New Britain (Xu et al., 2020) trenches, Xu et al. (2021), using
elemental ratios of organic matter, proposed that the Atacama and Kermadec Trenches contain a larger
contribution of terrigenous organic matter. Given the labile nature of ester-bond eukaryotic and bacterial IPLs
(Lipp et al., 2008; Schouten et al., 2010; Logemann et al., 2011), our study indicates that indigenous organic
matter from *in situ* microbial production dominates the pool of labile organic carbon in sediments of the AT and
the bathyal region. However, since IPLs do not allow us to estimate the contribution of refractory organic matter
coming from allochthonous sources, further work on IPL degradation products and other lipid classes is warranted.

**4.7. Characteristic IPLs of hadal and bathyal sediments**
The IPLs that contribute most to the dissimilarity between the hierarchical cluster containing samples from the
hadal and bathyal sediments (Cluster 1 of Fig. 10) and the water column (cluster 2, 3, 4 and 5 of Fig. 10) are
represented in Fig. 11. The most characteristic IPLs of hadal and bathyal sediments are: DGCC-42:6, DGCC-
27:0, DGCC-26:0, PC-35:0, PC-30:1, PC-30:2, PC-33:2, PC-32:1, PC-29:2, PE-DAG-32:1, PE-DAG-32:2, PE-
DAG-33:1, PI-AR, PG-DAG-36:2, and DGDG-34:2, which we propose as potential markers for these
environments. Even though DGCCs have been mainly related to algae membranes (Kato et al., 1994; Van Mooy
et al., 2009), DGCCs were minor components of the water column in this area, suggesting a different source for
these IPLs. In addition to DGCC, the other two betaine lipids, DGTA and DGTS, exhibited five IPLs that were
almost exclusively present in sediment samples (DGTA-34:1, DGTA-32:1, DGTA-34:2, DGTS-34:0 and DGTS-
32:1, see Figure 11). We note that almost all the PC phospholipids in our study have not, to the best of our
knowledge, been previously reported in the literature, which reinforces their use as markers of *in situ* bathyal and
hadal production in sediments.



The presence of a few MGDGs and SQs in hadal and bathyal sediments indicates that at least some labile organic
matter could derive from the shallow water column (see section 4.3). However, the most abundant IPLs in our
sediment samples, DGCC-42:6, PC-35:0, PI-AR, PE-DAG-32:1 and PG-DAG-36:2 (Fig. S7), are almost
completely absent in the overlying water column (Fig. 11). This reinforces the idea that these IPLs most likely
originate from *in situ* microbial production in sediments.

The most abundant IPL in sediments, DGCC-42:6, was not present in cluster 1, which only contains hadal
sediments (Figs. 2 and 3). Instead, this compound is prominent in clusters 3, 4, and 5, which contain both hadal
and bathyal samples. Thus, this IPL appears to be an indicator of lateral transport from bathyal regions, as could
be PC-35:0, which has the lowest relative abundance in the cluster only containing hadal sediments. The other
highly abundant IPLs in sediment and the hadal cluster (cluster 1 in Fig. 3), PI-AR, PE-DAG-32:1 and PG-DAG-
36:2, are also likely indicators of *in situ* microbial biomass. Finally, we acknowledge that temporal variability in
IPL production in the upper water column could complicate some of these assignments. For example, while PG-
DAG-36:2 was not present in the overlying water column of the Atacama Trench (Fig. 11), it has been previously
related to heterotrophic bacteria in surface waters of the South Pacific Ocean (Van Mooy and Fredricks, 2010).
Thus, future studies on the temporal variability of IPLs produced in the water column (and sediments) would
strengthen some of the associations we see in our data.


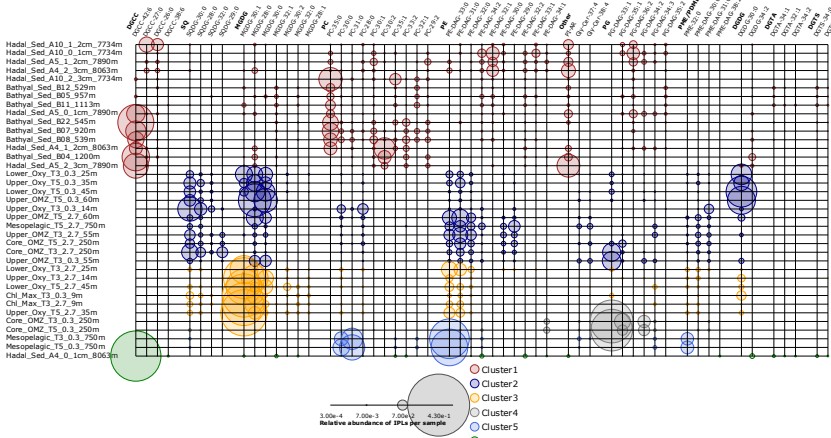

**Figure 11. Relative abundance of individual IPLs that contribute most to the dissimilarity between clusters of Fig. 10**
**derived from the SIMPER analysis (Table S1). Circle ratios are proportional to the relative abundance of IPL**
**compounds per sample. Samples are organized along the Y axis and shown in colors that match the hierarchical cluster**
**analysis in Fig. 10. The scale for circumference size is shown in the legend.**




### 4.8 Do IPLs reveal homeoviscous adaptation to the deep-sea environment?

Environmental factors such as pH, conductivity, temperature, and pressure impact the permeability and fluidity of cell membranes (Shaw, 1974; Macdonald, 1984; DeLong and Yayanos, 1985; Somero, 1992; Komatsu and Chong, 1998; Van Mooy et al., 2009; Carini et al., 2015; Sebastián et al., 2016; Siliakus et al., 2017; Boyer et al., 2020). Thus, organisms adapt to changes in environmental factors to maintain physiological homeostasis by altering their fatty acid composition (DeLong and Yayanos, 1985; Fang et al., 2000; Nichols et al., 2004; Siliakus et al., 2017). The combined physiological effects of high hydrostatic pressure and low temperature on prokaryotic membranes in laboratory cultures leads to the production of unsaturated lipids (DeLong and Yayanos 1985; Fang et al., 2000; Nichols et al., 2004; Zheng et al., 2020). However, few studies have been conducted using culture-independent techniques in search for potential adaptation mechanisms in organisms inhabiting the deep ocean (i.e., Zhong et al., 2020). We sought to understand whether the chemical composition of core fatty acids within different IPL classes (i.e., carbon length and unsaturation degree) reflects the combined effects of the low temperature and high pressure typical of hadal settings. We show that PGs are abundant in hadal sediments of the AT (Fig. S4). Fang et al., (2000) using strains isolated from Mariana Trench sediments, found that PG was the most abundant class of phospholipids and presumed that it could be a physiological response to high pressure and low temperature, subsequent studies have confirmed this approach (Winter et al., 2009; Periasamy et al., 2009; Jebbar et al., 2015, Allemann et al., 2021). Cluster 1 in the boxplot analysis (Fig. 4) likely contains the most characteristic IPL classes of the hadal zone. In general, the phospholipid class in this cluster exhibited comparatively fatty acid chains that are monounsaturated and saturated compared to other environments (Figs. 4a, b). Additionally, we observed an increase in the ratio of total unsaturated to saturated fatty acids in deep sediments compared to the water column (Fig. 5), which could reflect physiological adaptations of their biological producers. These results are in accord with studies indicating biosynthesis and incorporation of polyunsaturated fatty acids into phospholipid membranes of piezophilic bacteria (DeLong and Yayanos, 1985; Baird et al., 1985; Yano et al., 1998; Winter, 2002; Mangelsdorf et al., 2005; Winter and Jeworrek, 2009; Allemann et al., 2021). Thus, the chemical characteristics (C length and degree of unsaturation) of the most abundant IPLs in sediments of the AT suggest homeoviscous adaptation to this type of environment by their source organisms, in addition to potentially indicating the occurrence of compounds that are unique to the endogenous community.

### 5. Conclusions

IPLs in surface hadal sediments from the deepest points of the Atacama Trench share characteristics with those in bathyal sediments and differ from those found in particles from the upper ocean, including the euphotic zone and the oxygen minimum zone. This suggests that IPLs in the hadal region are most representative of lateral sediment transport from shallower depths combined with *in situ* microbial production.

The most dominant IPL structures in the bathyal and hadal sediments show a great variety of phospholipids with varying degrees of unsaturation that are likely bacterial in origin. Hadal sediments also exhibit unique glycolipid structures such as SQDG-42:11, SQDG-23:0, DGDG-35:1, DGDG-35:2 and DGDG-37:1 that have not, to the best of our knowledge, been reported in other environments. Although, they are present in low abundance and represent a small fraction (~0.00012%) of the total IPL pool in sediments. Furthermore, hadal sediments exhibited high spatial heterogeneity in IPL distribution. The similarities in IPLs between hadal and bathyal sediments

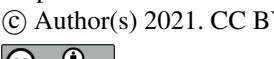



supports lateral transport of organic matter or the occurrence of potentially similar *in situ* bacterial communities
with traits such as mono-unsaturated and high ratios of unsaturated/saturated fatty acids that possibly indicate
homeoviscous adaptation to the high pressure and low temperatures found in deep-sea environments.

An improved understanding of the phylogenetic and metabolic association of IPLs, as well as their ecological role
in this unique and extreme environment, could be achieved in future studies by including a detailed analysis of
the biogeochemical conditions as well as the composition of the microbial community in hadal sediments from
the Atacama trench.

**Author contribution**


EF, OU, and JS designed the study. MZ contributed with the hadal samples from the HADES-ERC cruise. EF
prepared, extracted, and analyzed samples from the HADES-ERC cruise with help from SC and ND under the
supervision of JS. EF and SC processed results. EF, SC, and JS interpreted results. EF and PR-F performed
statistical analyses. EF wrote the manuscript with contributions from SC, JS, and OU. All authors provided
feedback on the manuscript. OU and JS funded the research.

**Competing interests**


The authors declare that they have no conflict of interest.

**Acknowledgements**


This work was supported by the Chilean Agency for Research and Development (grants ICN12_019-IMO and
FONDECYT 1191360 to O. Ulloa). Additional support was provided by the Department of Geological Sciences
and INSTAAR at the University of Colorado Boulder (to J. Sepúlveda), the European Research Council (Hades-
ERC, grant agreement number 669947, to R.N. Glud), and the Max Planck Society. EF was also partially
supported by the UCO 1866 Student Scholarship-2019, Directorate of Graduate Studies, Universidad of
Concepción. We are thankful to the captains, crews, and scientists of the German RV *Sonne* cruises SO-261
(HADES-ERC) and SO-211 (ChiMeBo). In particular, we thank the chief scientists R.N. Glud and F. Wenzhöfer
(HADES-ERC) and D. Hebbeln (ChiMeBo). The HADES-ERC and ChiMeBo cruises were funded by the
European Research Council and the German Bundesministerium für Bildung & Forschung (BMBF),
respectively. We also wish to thank Carina Lange and Silvio Pantoja for access to samples from the ChiMeBo
cruise and M. Pizarro-Koch for the preparation of the three-dimensional map. We also wish to thank L. Nuñez,
B. Srain, R. Castro, A. Ávila, M. Mohtadi, R. De Pol-Holz, and G. Martínez-Méndez, for sample collection during
the ChiMeBo cruise and/or laboratory assistance.






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
