# Peer review of "Bacterial and eukaryotic intact polar lipids point to *in situ* production as a key source of labile organic matter in hadal surface sediment of the Atacama Trench"

_Biogeosciences, 2021_

## Author Response (AR1)

Response to Reviewer 1 Comments (RC1):

**We thank the reviewer for the constructive comments**

Specific comments

RC1 – 1: The lipid extraction and analytical methods are appropriate for the sample types in the study. Please report the type of chromatographic column in the methods section.

**Response RC1 – 1: We will add details of the chromatographic column in the methods section:** *"We used Waters Acquity BEH Amide column (2.1 × 150 mm; 1.7 µm particle size) that enables class-specific separation of polar lipids based on their hydrophilic head group (Wörmer et al., 2013)."*

RC1 – 2: I appreciate the use of deuterated standards to account for matrix effect on quantifying lipids. Would it be possible to include an assessment of the importance of this treatment for the benefit of planning future studies on sedimentary IPLs? How was the correction applied? What was the overall impact of including this step on the final data reported? Were there different effects observed for the different lipid classes? Since deuterated standards were added just for PC, PG, PE and DGTS, was there a way to correct the concentration calculations for the other lipid classes as well? If not, how does this affect quantitative comparisons between lipid classes?

**Response RC1 – 2: We agree that this is an important aspect to highlight in more detail for future studies. We corrected for the matrix effect on ionization efficiency by comparing the loss of signal observed between deuterated standards (PC, PG, PE and DGTS) analyzed both pure and spiked to samples. While we did not have deuterated standards available for other IPL classes, we found that the matrix effect was consistent among the four standards we tested and accounts for a ~7±0.6 % loss in ionization efficiency. Therefore, since the matrix effect appears to have a similar impact on these 4 different lipids classes, our overall results are unlikely to be significantly biased by the lack of other deuterated standards. We plan to acknowledge these observations more explicitly in the revised version of the manuscript.**

*"Unfortunately, we could not find more deuterated standards to test for other IPL classes. However, on average, we observed that the matrix effect accounts for a ~7±0.6% loss in ionization efficiency for the four available deuterated standards. Therefore, it is reasonable to assume a similar loss for other IPL classes, although this remains to be tested in future studies. We highlight the importance of using as many*

*IPLs classes as possible to account for both differences in ionization and matrix effect when performing IPL quantification in environmental samples."*

RC1 – 3: Figure 3 is very helpful for understanding the clusters described in Figure 2. It shows which specific IPLs are the main controls on the differences in lipid class composition that defines clusters. The font size in figure 3 is very small, though, so could that figure be rotated 90 degrees to increase its size?

**Response RC1 – 3: We will correct this figure in the revised version.**

RC1 – 4: In Table 2 and in the text, how did the authors decide to use the DAG designation for PE and PG abbreviations at some places and not others? PC also has AEG species, so should PC-DAG be used? It also appears that SQ and SQDG are used interchangeably. I'm mostly thinking about using consistent naming, so the reader isn't looking for a distinction between different abbreviations used for the same compound.

**Response RC1 – 4: Thank you for this remark. We use DAG to designate a diacylglycerol and AEG to designate an acyletherglycerol; thus, we will use this same nomenclature for PCs. Also, we refer to SQ throughout the manuscript as SQDG. We will standardize all compound abbreviations to ensure consistency.**

RC1 – 5: In sections 4.2, 4.3, and 4.4, I really like the introduction paragraphs for each IPL class describing possible biological sources.

**Response RC1 – 5: We are glad to hear that the inclusion of this text is meaningful.**

RC1 – 6:  How do the authors define IPLs with short vs. long-chain fatty acids? Is there are consistent definition, or does it vary by IPL class? In lines 472-473 and 575-577 there is overlap in the ranges.

**Response RC1 – 6: We describe carbon ranges following previous studies (i.e. Rêzanka et al., 2009; Schubotz et al., 2009a; Brandsma., et al., 2011), where short- and long-chains refer to combined alkyl chains of $C_{28}$-$C_{36}$ and $C_{36}$-$C_{44}$, respectively.  We will add this information in the revised version.**

RC1 – 7: Figures 7, 8,and 9 are very difficult to read at the size of a printed page. Would it be possible to provide a simpler summary figure in the manuscript and include the three figures as nine individual figures in the supporting information?

**Response RC1 – 7: We agree with the reviewer about the benefits of presenting a simpler summary of these three different figures while moving their detailed**

**individual versions to the supporting information. We will change this in the revised version.**

RC1 – 8: The conclusions section could better represent the work. Conclusion 1 (801-804) is well supported. The middle part of conclusion 2 (806-814) repeats part of conclusion 1. The five IPLs listed on line 808 are in such low abundance that I don't think they warrant their own conclusion as possible distinctive biomarkers of trench communities. They could very well be in other biota but unreported because they are quantitatively insignificant in an organism's lipidome. Why not focus conclusions more on drawing specific conclusions based on the significant differences in PG, PC, MGDG, DGDG, SQDG, DGCC, DGTS, and DGTA that are so clear in Figures 7-9? That's really interesting and should be a major part of the conclusions.

**Response RC1 – 8: We agree with the referee's comment regarding a revised conclusions section that better highlights the most important contributions of our work. We will add this in the revised version. Please see a more detailed reply to a similar comment by Reviewer 2 below (RC2 – 4).**

Technical Corrections:

RC1 – 9: Line 596: Add heading for Potential Sources of Betaine Lipids.

**Response RC1 – 9: We will correct this.**

RC1 – 10: Line 649: Are the 21-38 carbons just on the fatty acid or also on the ceramide chain?

**Response RC1 – 10: We report both, the total number of carbons including the ceramide chain and the fatty acids. We will add this sentence:** *"[...] shows Gly-Cer with ceramide chain, and polyunsaturated fatty acids with $C_{21-38}$ [...]"*

RC1 – 11: Line 694-696: Correct incomplete sentence.

**RC1 – 11: We will correct this in the revised version.**

Response to Reviewer 2 Comments (RC2):

**We thank the reviewer for the constructive comments**

General comments

RC2 – 1: This paper is generally well written (more so for the results and discussion). The data represent an important contribution.

**Response RC2 – 1: Thank you for the overall positive review of our work.**

RC2 – 2: Although, I am in favor of publication, I do not yet necessarily agree with the interpretation or rationalization of the described lipidome. From the paper's title to its conclusion is a focus of IPLs largely representing (or acting as) tracers or proxies of particulate organic matter (POM). While a component of IPLs may find its way into POM, these compounds say little about what POM is, how it comes to be preserved, or how it is transported through the water column. The OM instead comprised of all sorts of detritus (i.e. extracellular polysaccharides, fecal material, animal kills and falls, etc.).

**Response RC2 – 2: Thank you for this remark and for the opportunity to clarify the scope of our work. We are aware of the complex composition of POM, its transformation along its transit through the water column, and the fact that IPLs only represent a fraction of it. We will highlight this aspect more explicitly in the revised version of the manuscript. Given the labile nature of ester-bond IPLs (Logemann et al., 2011), however, they are widely used as tracers of in situ living biomass or recently dead microbial cell material (White et al., 1979; Harvey et al., 1986; Petersen et al., 1991). Thus, the main scope of our work is to determine whether the most labile fraction of the lipid pool present as IPLs in deep-sea sediments of the Atacama Trench derive from in situ sources or a combination of in situ sources and the water column. The only aspect of POM transformation that we can discuss is that a neglectable fraction of the IPLs produced in the upper 750 m of the overlying water column reported by Cantarero et al. (2020) is present in surface sediments of the bathyal and hadal regions of the Atacama Trench. This result implies a rapid IPL degradation during sinking, which is consistent with experimental degradation rates (Westrich and Berner, 1984; Logemann et al., 2011) and first-order POM sinking rates.**

RC2 – 3: Furthermore, the authors explain the similarities between the bathyal and hadal lipidomes as products of mass sediment transport, what they term as being lateral transport. First the term lateral transport is confusing as this could easily also mean transport across on latitudinal position within the base of the trough to another point at the base of the trough. Instead, I think the authors are referring to down slope

sedimentation.  Second, the basis for a lateral transport does not to me make sense based on the data that have been collected. Down-slope mass wasting events are episodic in space and time (even the V-shaped basins vary in slope from 8-10deg). The sediment depths to which the samples were collect are at the very surface layers (0-1cm deep and no more than 3 cm deep).  So, if the kinetics show that upper water column inputs do not survive their direct transport to the seafloor, then how does an IPL survey as a sediment particle moving step-wise down the trench slope? What sedimentary evidence do the authors have that the upper 3 cm of sediment at all of the sampled stations represents debris flows, turbidite, or mass wasting events? What is the slopes adjacent to all AT core sites? Can a model be made to show what the decent time would be for a bathyl sediment to reach the bottom of the trench?

**Response RC2 – 3: We thank the reviewer for this excellent point and for the opportunity to clarify this aspect. First, we agree that the term "lateral transport" is confusing, and that "downslope transport" is more appropriate to explain the potential transfer of sediment from the bathyal to the hadal region. Second, we also agree that we are not able to distinguish between lateral transport within the Atacama Trench from downslope transport from the bathyal region. Third, the lack of sedimentological and geochemical data, in addition to the spatial and temporal limitations of our study, a sediment transport model for the study area is far beyond the scope of our study and warrants further investigation.**

**Thus, given the main focus of our study and the lability of IPLs, we will reduce the role of these mechanisms in our revised manuscript to a hypothesis to be tested by future studies.**

**Our revised discussion will include the role of tectonic processes, including recently published and/or ongoing work that we were not aware of before the submission of our manuscript. Recent work in the Japan trench indicates that down deposits arise from earthquake-driven remobilization of surface sediments from the continental shelves (Schwestermann et al., 2021). We are now also aware of similar studies currently in preparation for the Atacama and Kermadec trenches (Stewart et al., *in prep*.; Chen et al., *in prep.;* Zabel et al., *submitted*), which we plan to mention. While we lack sedimentological/geochemical data to discriminate if the top 3 cm of our hadal stations A4, A5 and A10, represent debris flows, turbidite, or mass wasting events, a study currently under review addresses this question (Oguri et al., submitted). These authors use $^{210}Pb_{ex}$ measurements to indicate that stations A4 and A5 exhibited mono exponential declines in the $^{210}Pb_{ex}$ from the sediment surface until depths were they reach low background levels, suggesting no mixing or mass wasting at both**

**sites in the past ~160-180 years. Similarly, other sites of the Atacama Trench indicate that surface sediments consist of continuous hemipelagic sedimentation, except for Station A10, which seems to have been formed by a recent turbidity layer. We plan to place our results within this sedimentological framework to draw revised discussion and conclusions sections.**

RC2 – 4: More likely is that the microbial communities of the sea floor are not sensitive to the change in hydrostatic pressure and all things the same, thrive on a low nutrient supply from the upper water column – creating its own selective mechanism. As such the paper would be better served focusing on the important conclusions that it does resolve well. These are:

1.  This is an environmental baseline survey of bacterial and eukaryotic sourced IPLs.

2.  The most abundant ester-based IPL are phospholipids. Most of these have yet to be described.

3.  Bathyal and Hadal sediments have very similar compositions of ester-bound IPLs and therefore may indicate that these environments are host to the same microbial surface communities.

4.  Most IPLs that would be common to the upper water column appear to get almost entirely degraded during their descent to the hadal seafloor suggesting the highly labile lipids are derived from ocean floor microbial communities.

**Response RC2 – 4: We thank the reviewer for this valuable comment. We plan to rewrite the conclusions section to more effectively summarize the key findings of our study following the draft below.**

1.  **Bacterial and eukaryotic sourced IPLs in surface hadal sediments from the deepest points of the Atacama Trench share characteristics with those in bathyal sediments and differ from those found in suspended particles from the upper 750 m of the water column, including the oxygen minimum zone. This indicates that: a) most IPLs abounding the upper water column are almost entirely degraded during their descent to the hadal seafloor, and b) IPLs found in hadal sediments are predominantly derived from *in situ* microbial communities.**

2.  **The most dominant ester-bound IPL structures found in bathyal and hadal sediments show a great variety of phospholipids with varying degrees of unsaturation, most of them yet to be described, that are likely bacterial and/or fungal in origin. Hadal sediments also exhibit unique glycolipid structures, such as SQDG-42:11, SQDG-23:0, DGDG-35:1, DGDG-35:2 and**

**DGDG-37:1, that have not yet, to the best of our knowledge, been reported in other environments. However, these lipids are present in low abundance and represent a small fraction (~0.00012%) of the total IPL pool. Elevated ratios of unsaturated/saturated fatty acids in hadal sediments are likely indicative of homeoviscous adaptation to the high pressure and low temperatures characteristic of this extreme deep-sea environment.**

3. **An improved understanding of the phylogenetic, ecological, and metabolic association of IPLs present in the Atacama Trench could be achieved in future studies by the pairing of lipidomics with genomic techniques (e.g., microbial community composition, functional groups, lipid biosynthesis) in addition to a detailed sedimentological and biogeochemical characterization of sediments.**

RC2 – 5: Lastly, I would suggest that some of the eukaryotic IPLs may represent fungi or metazoan type detritivor. That would be a interesting use of IPLs if this link could be constrained.

**Response RC2 – 5: We thank the reviewer for this suggestion. We will add a discussion about potential fungal and/or metazoan sources, including information from the recent publications listed below.**

**Gao, Y., Du, X. Xu, W., Fan, R., Zhang, X., Yang, S., Chen, X., Lv, J., Luo, Z. Fungal diversity in deep sea sediments from east yap trench and their denitrification potential. Geomicrobiol. J., 1–11, 2020.**

**Gutiérrez, M. H., Vera J., Srain B., Quiñones, R. A., Wörmer L., Hinrichs K.-U., and Pantoja, S. Biochemical fingerprints of marine fungi: implications for trophic and biogeochemical studies. Aquat. Microb. Ecol., 2020.**

Minor Comments

RC2 – 6: Title: Archaeal lipids do not factor into this study, so perhaps indicate this is a bacterial and eukaryote IPL survey. The choice of indicating labile organic matter is not something I think gives strength to this study. Please see above comments for more on this point.

**Response RC2 – 6: Thank you for the suggestion. We plan to use an updated title like the one below.**

**"Bacterial and eukaryotic intact polar lipids point to in situ production as a key source of labile organic matter in hadal surface sediment of the Atacama Trench"**

RC2 – 7:  Consider removing lateral transport.

**Response RC2 – 7: As explained above, we will replace that term by "downslope transport" in the revised version of the manuscript.**

Abstract:

RC2 – 8:  Line 20 – 21: Change sentence to "Elevated organic matter (OM) concentrations are found in hadal surface sediments relative to the surrounding abyssal seabed. However, the origin of the biological material remains elusive.

RC2 – 9: From here on replace all instances of the term "organic matter" with the acronym "OM".

RC2 – 10: Line 22: Replace "cell" with "cellular" and "in" with "extracted from surface sediments".

RC2 – 11: Line 23: replace "depths" with "margin".

**Response to RC2 8-11: We will include these suggestions.**

RC2 – 12: Line 26 – 28: Unclear, please rewrite.

**Response RC2 – 12: We will modify this in the revised manuscript.**

RC2 – 13: Line 29: Delete labile – all IPLs are labile lipid structures with some head group classes being more resilient than others.

**Response RC2 – 13: We will include this suggestion.**

RC2 – 14: Line 29 – 30: Does not fall out as to how that is necessarily so based on what is written.

**Response RC2 – 14:  We will modify this in the revised version.**

RC2 – 15: Line 35: End sentence at ecosystem. Begin the next sentence with Furthermore, they also…

**Response RC2 – 15: We will include this suggestion.**

RC2 – 16: The abstract does little to reconstruct the microbial diversity based on the recovered lipidomes as extensively discussed in the text.

**Response RC2 – 16: We will modify this in the revised version.**

Introduction

RC2 – 17:  Line 41: delete "ocean".

RC2 – 18: Line 42: delete "long-held".

RC2 – 19: Line 44: Replace "In" with "For" and delete "while".

RC2 – 20: Line 45: add "additionally" after pressure.  And delete "the most" end sentence by deleting "compared to shallower habitats.

RC2 – 21: Line 46: Begin a new sentence with "Availability" and move this sentence up in front of the prior one.

RC2 – 22: Line48: Replace "However" with "To"

RC2 – 23: Line 49: Delete "A study by".

RC2 – 24: Line 54: Use POM.

**Response to RC2 17-24: We will include these suggestions in the revised version.**

RC2 – 25: Line 59: I would argue that all sources of OM spatially vary.

**Response to RC2 – 25:  We will modify this sentence: "** *these are spatially and temporally variable* **"**

RC2 – 26: Line 60: Unclear. This is an over-simplification of the mass transport mechanisms and hydrographic processes at play in trench systems.

**Response RC2 – 26: We will modify this sentence to better reflect the complexity of sediment transport processes in marine trenches**

RC2 – 27: Line 62: Replace "bacterial and archaeal" with "microbial". Please also note that these surface sediments certainly contain abundant fungal communities.

**Response RC2 – 27: We will modify this sentence accordingly.**

RC2 – 28: Line 64: Missing references.

**Response RC2 – 28: The following additional reference will be included: Grabowski, E., Letelier, R. M., Laws, E. A., and Karl, D. M.: Coupling carbon and energy fluxes in the North Pacific Subtropical Gyre, Nat. Commun., 10, 1895, 2019.**

RC2 – 29: Line 67: Rewrite this section of the sentence.

**Response RC2 – 29: We will modify this sentence as *"the benthic oxygen consumption can vary significantly in hadal sediments (Glud et al., 2021)."***

RC2 – 30: Line 68: It is unclear what is really meant by the term redistribution in this sentence.

**Response RC2 – 30: We will modify this sentence as: *"can be influenced by dynamic depositional conditions (Schauberger et al., 2021), …"***

RC2 – 31: Lines 82 – 85: Too much detail.

**Response RC2 – 31: We plan to simplify this sentence in the revised version of the manuscript.**

RC2 – 32: Lines 85 – 86: Not only. Other head groups are common as well.

**Response RC2 – 32: We will modify this in the revised version.**

RC2 – 33: Figure 1. Could be greatly improved with a larger perspective map showing where along the Atacama Trench the samples survey occurred.

**Response RC2 – 33: We will incorporate this helpful suggestion.**

RC2 – 34: Line 90: Under these conditions, it is not the glycerol or acyl/isoprenoid tails that are labile, but the head groups of the lipid. Also I am not seeing the point of this sentence. How can you compare sources and preservation based on to higher break-down products? Much of these CLs form similar break-down products that cannot in themselves be untangled and linked to their primary sources.

**Response RC2 – 34:  Please see our response to RC2 – 2**

RC2 – 35: Lines 102: Many things make up labile compounds that are not IPLs.

**Response RC2 – 35:  We plan to be more explicit and indicate that IPLs represent only one of the several labile fractions of OM.**

RC2 – 36: Line 107: It is a mischaracterization to consider IPLS as a proxy of OM loading to sediments. Microbes may hitch-hike on other forms of detritus (fecal material, extracellular polysaccharides, clay minerals, etc.), but they easily represent (and most commonly and simply do represent) what is the living or very recently deceases microbial components in the sediment. The very nature of applying a modified Bligh and Dyer extraction to get IPLs insinuates that these compounds are still attached as complete or partially degraded cellular membranes. If OM provenance is to be assessed from IPLs it must be done in combination with traditional techniques, be they hydrocarbon biomarker analyses, bulk rock sediment parametrization (TOC, HI, and OI) and/or FT-ICRMS POM studies.

**Response RC2 – 36:  We agree with the reviewer that other biomarkers/analyses are necessary to evaluate the origin of organic matter in more detail, which is beyond the specific scope of our study, but part of planned investigations. We will modify the text and be more explicit about our interpretation of IPL signatures in sediments. In a nutshell, and as explained above, we use them as indicators of in situ microbial production and to demonstrate that IPLs produced in surface waters degrade through their transit to the seafloor.**

Materials and Methods

RC2 – 37: Line 120: Indicate that this is bacterial and eukaryote IPLs.

**Response RC2 – 37: We will add this.**

RC2 – 38: Line 121: Delete "with 3 depth intervals each. This is better explained below.

**Response RC2 – 38: We will modify this in the revised version.**

RC2 – 39: Line 144: It is unclear why on three very shallow sediment samples (up to 3 cmbsf) were analyzed within a 60 cm core.

**Response RC2 – 39: Given the lability of IPLs, we focused on the top 3 cm of the sediment column to facilitate their comparison with surface sediment samples from the bathyal region as well as with POM from surface waters. We will provide an expanded justification for this rationale in the revised version of the manuscript.**

RC2 – 40: Section 2.2.1 Lipid extraction – Comment: Presumably the samples were immediately sectioned and frozen when removed from the multicorer?  No details provided on this stage of sample collection and processing.

**Response RC2 – 40: Yes, they were. We will add more details about the sampling, processing, and storing of samples.**

RC2 – 41: Lines 164 – 176: Make lower case the chemical names that do not start a sentence. Continue to follow this throughout the text.

**Response RC2 – 41: We will modify this in the revised version.**

Results

RC2 – 42: Line 262 – 267: Acronyms already defined in the text.

RC2 – 43: Line 385: Replace "test" with "evaluate".

**Response to RC2 42-43: We will incorporate these suggestions.**

Discussion

RC2 – 44: Lines 429 – 431: Not needed and can be deleted.

RC2 – 45: Lines 717 – 722: Simplify this section.

RC2 – 46: Lines 723 – 724: Delete this sentence.

**Response to RC2 44-46: We will incorporate these suggestions.**

RC2 – 47: Line 741: Change to SQDG?

**Response RC2 – 47: We will modify this and provide a more standardized nomenclature of IPL structures (see response to RC1 – 4).**